# Maternal obesity blunts antimicrobial responses in fetal monocytes

Suhas Sureshchandra[1,2], Brianna M Doratt[2,3], Norma Mendza[2], Oleg Varlamov[4], Monica Rincon[5], Nicole E Marshall[5], Ilhem Messaoudi[2,5]*

[1]Institute for Immunology, University of California, Irvine, Irvine, United States; [2]Department of Molecular Biology and Biochemistry, University of California, Irvine, Irvine, United States; [3]Department of Microbiology, Immunology, and Molecular Genetics, University of Kentucky, Lexington, United States; [4]Division of Cardiometabolic Health, Oregon National Primate Research Center, Oregon Health & Science University, Beaverton, United States; [5]Maternal-Fetal Medicine, Oregon Health & Science University, Portland, United States

**Abstract** Maternal pre-pregnancy (pregravid) obesity is associated with adverse outcomes for both mother and offspring. Amongst the complications for the offspring is increased susceptibility and severity of neonatal infections necessitating admission to the intensive care unit, notably bacterial sepsis and enterocolitis. Previous studies have reported aberrant responses to LPS and polyclonal stimulation by umbilical cord blood monocytes that were mediated by alterations in the epigenome. In this study, we show that pregravid obesity dysregulates umbilical cord blood monocyte responses to bacterial and viral pathogens. Specifically, interferon-stimulated gene expression and inflammatory responses to respiratory syncytial virus (RSV) and *E. coli* were significantly dampened, respectively . Although upstream signaling events were comparable, translocation of the key transcription factor NF-κB and chromatin accessibility at pro-inflammatory gene promoters following TLR stimulation was significantly attenuated. Using a rhesus macaque model of western style diet-induced obesity, we further demonstrate that this defect is detected in fetal peripheral monocytes and tissue-resident macrophages during gestation. Collectively, these data indicate that maternal obesity alters metabolic, signaling, and epigenetic profiles of fetal monocytes leading to a state of immune paralysis during late gestation and at birth.

## Editor's evaluation

This manuscript will be of interest to a broad range of researchers studying immunology, obesity and metabolism, links between maternal health and pathophysiological responses in the offspring. The comprehensive studies using RNA-seq, scRNA-seq, ATAC-seq, scATAC-seq and CUT& Tag represent an important resource for understanding the transcriptomic and epigenetic shifts in the monocytes of newborns. The experiments involving stimulation of monocytes with pathogens offer convincing evidence for the dysfunction of monocytes in the newborn. The analysis of fetal macrophages of non-human primates are of fundamental importance for the field.

## Introduction

Nearly 40% of women of childbearing age (18–44 years old) in the United States met body mass index (BMI) obesity criteria in 2019 (*Wang et al., 2020*) and the rate of obesity amongst women of childbearing age continues to rise with ~11% increase in prevalence from 2016 to 2019 (*Driscoll and Gregory, 2020*). Consequently, pre-pregnancy (pregravid) obesity has emerged as one of the leading

*For correspondence:
ilhem.messaoudi@uky.edu

Competing interest: The authors declare that no competing interests exist.

co-morbidities that impact maternal and fetal health. The Developmental Origins of Health and Disease (DOHaD) concept (*Barker et al., 1989b*; *Barker et al., 1989a*) proposes that both maternal overnutrition and undernutrition during fetal development can have significant consequences on the offspring's health during childhood and adult life (*Bansal and Simmons, 2018*; *Friedman, 2018*). Indeed, it is now well established that pregravid obesity is associated with adverse outcomes for both mother and offspring (*Lashen et al., 2004*; *Stothard et al., 2009*). Compared to lean mothers, mothers with obesity have a substantially increased risk of developing gestational diabetes, pre-eclampsia, and hypertension all associated with small for gestational age neonates and increased risk of stillbirth (*Yao et al., 2017*; *Howell and Powell, 2017*). Pregravid obesity is linked to an increased risk of necrotizing enterocolitis, sepsis, and severe respiratory syncytial virus (RSV) infection during the neonatal period necessitating admission to the intensive care unit (*Griffiths et al., 2016*; *Rastogi et al., 2015*; *Suk et al., 2016*). Additionally, maternal obesity is associated with a higher incidence of long-term consequences for the offspring including increased incidence of allergies, asthma, and metabolic disorders (*Catalano, 2007*; *Gaillard et al., 2014*; *Gupta et al., 2022*; *Wilson et al., 2015*; *Ardura-Garcia et al., 2021*; *Godfrey et al., 2017*; *Sharp et al., 2017*; *Briese et al., 2010*; *Castro and Avina, 2002*; *Danieli-Gruber et al., 2017*; *American College of Obstetricians and Gynecologists' Committee on Practice Bulletins–Obstetrics, 2021*).

Similarly, studies using animal models where maternal obesity is induced via the administration of high fat or western style diets (HFD, WSD) showed that pups born to obese dams fed a HFD during gestation generated lower ovalbumin (OVA)-specific IgG, but higher OVA-specific IgE responses, after immunization with OVA compared to pups born to dams fed a control diet (*Odaka et al., 2010*). These data potentially explain the increased incidence of asthma/wheezing in children born to mothers with obesity (*Gaillard et al., 2014*). Further studies show that pups born to obese dams fed a HFD are more susceptible to bacterial challenge (*Myles et al., 2013*), develop more severe bronchiolitis following RSV challenge (*Griffiths et al., 2016*), and are more likely to develop autoimmune encephalitis (*Myles et al., 2013*). Moreover, ex vivo LPS stimulation of colonic lamina propria lymphocytes (LPLs) isolated from pups born to dams fed a HFD resulted in greater levels of IL-6, IL-1β, and IL-17, whereas LPS stimulation of splenocytes from the same animals generated decreased levels of TNFα and IL-6 compared to control groups (*Myles et al., 2013*).

These observations suggest that pregravid obesity disrupts the development and maturation of the offspring's immune system in utero. Indeed, clinical studies showed that high pregravid BMI is associated with reduced frequency of umbilical cord blood (UCB) CD4+ T cells and dampened CD4+ T cell responses to polyclonal stimulation (*Wilson et al., 2015*; *Sureshchandra et al., 2021c*; *Sureshchandra et al., 2017*; *Gonzalez-Espinosa et al., 2016*). Additionally, pregravid obesity was associated with a reduced ability of human UCB CD14+ monocytes to respond to ex vivo stimulation with toll-like receptor (TLR) ligands (*Wilson et al., 2015*; *Sureshchandra et al., 2017*). Moreover, monocytes from offspring of mothers with obesity showed a decrease in transcripts of the pro-inflammatory cytokines IL-1ß and IL-12ß (*Cifuentes-Zúñiga et al., 2017*). These dampened CD4+ T cell and monocyte responses are mediated by epigenetic changes at key loci that regulate inflammatory responses and cellular differentiation pathways (*Sureshchandra et al., 2021c*; *Sureshchandra et al., 2017*). Other studies have reported an increased expression of PPAR-γ, which can attenuate macrophage inflammatory responses (*Chawla, 2010*) in peripheral blood mononuclear cells (PBMCs) obtained from children born to mothers with obesity (*Gaytán-Pacheco et al., 2021*). Similarly, we reported hypomethylation of the PPAR-γ gene in UCB monocytes obtained from mothers with obesity, thus indicating sustained reprogramming (*Sureshchandra et al., 2017*). Alterations in the DNA methylation patterns of the promoter regions in monocytes of the obese group suggest dysregulated responses to M1 and M2 macrophage polarization stimuli (*Sureshchandra et al., 2017*).

In this study, we extended these earlier clinical studies and interrogated the impact of pregravid obesity on the response to bacterial and viral pathogens. Our data showed dysregulated response by UCB monocytes from babies born to mothers with obesity. Additional analyses showed that upstream signaling events were comparable between UCB from lean and obese mothers; however, reduced chromatin accessibility and lack of chromatin remodeling in response to stimulation resulted in a dampened transcriptional response. This dampened response was also detected in peripheral blood monocytes and tissue-resident macrophages obtained from neonatal rhesus macaque born to obese dams fed a high-fat diet indicating that these observations are not limited to UCB monocytes.

**Table 1.** Cohort characteristics.

| | | Lean | Obese | Stats |
|---|---|---|---|---|
| * Enrolled | | 43 | 36 | |
| Maternal age at delivery (years) | | 33.1±4.4 | 30.7±4.9 | 0.058* |
| Pre-pregnancy BMI (kg/m²) | | 21.9±1.7 | 38.2±8.4 | <0.0001* |
| Gestational age at delivery (weeks) | | 39.6±1.4 | 39.3±1.4 | 0.374* |
| Fetal sex (Female %) | | 39.5 | 38.9 | 0.47[†] |
| Mode of delivery (%) | Cesarean | 30.23 | 55.55 | |
| | Vaginal | 62.79 | 25.00 | |
| | Unknown | 6.97 | 19.44 | 0.003[†] |
| Ethnicity (%) | Asian | 2.32 | 0.00 | |
| | Hispanic | 6.97 | 25.00 | |
| | Caucasian (non-hispanic) | 88.37 | 66.66 | |
| | More than one race | 0.00 | 2.77 | |
| | Unknown/Declined to state | 2.32 | 5.55 | 0.094[†] |

[*]Mann-Whitney T-test.

[†]Chi-squared test.

# Methods

## Subjects and experimental design

This study was approved by the Institutional Ethics Review Board of Oregon Health and Science University and the University of California, Irvine. Written consent was obtained from all subjects. The mean age of the 43 lean women was 33.1±4.4 years with an average pre-pregnancy BMI of 21.9±1.7 kg/m² and gestation age at delivery of 39.6±1.4 weeks; mean age of the 36 obese women was 30.7±4.9 years with an average pre-pregnancy BMI of 38.2±8.4 kg/m² and gestation age at delivery of 39.3±1.4 weeks (*Table 1*). The racial distribution of the lean cohort was 2.32% Asian American, 6.97% Hispanic, 88.37% Caucasian, and 2.32% declined to report. The obese cohort consisted of 25.00% Hispanic, 66.67% Caucasian, 2.77% with more than one race, and 5.55% who declined to report. Exclusion criteria included active maternal infection, documented fetal congenital anomalies, substance abuse, chronic illness requiring regular medication use, preeclampsia, gestational diabetes, chorioamnionitis, and significant medical conditions (active cancers, cardiac, renal, hepatic, or pulmonary diseases), or an abnormal glucose tolerance test. Women underwent a fasting blood draw and body composition via air displacement plethysmography using a BodPod (Life Measurement Inc, Concord, CA).

## Umbilical cord blood mononuclear cell (UCBMC) isolation

Complete blood counts were obtained by Beckman Coulter Hematology analyzer (Brea, CA) before cell isolation. Umbilical cord blood mononuclear cells (UCBMC) and plasma were obtained by standard density gradient centrifugation over Ficoll (GE Healthcare, Chicago, IL). UCBMC were frozen in 10% DMSO/FetalPlex (GeminiBio, Sacramento, CA) and stored in liquid nitrogen until analysis. Plasma was stored at –80 °C until analysis.

## Cord blood immunophenotyping

10[6] UCBMC were stained using the following cocktail of antibodies to enumerate innate immune cells and their subsets: PE-CD3, PE-CD19, PB-CD16, PE-Cy7-CD11c, AF700-CD14, PCP-Cy5.5-CD123, BV711-CD56, and APC-Cy7-HLA-DR. All samples were acquired with the Attune NxT Flow Cytometer (ThermoFisher Scientific, Waltham, MA) and analyzed using FlowJo 10.5.0 (Ashland, OR). Monocyte

subsets (classical, non-classical, and intermediate) were enumerated based on expression of CD14 and CD16.

## Intracellular cytokine assay

To measure cytokine responses of monocytes and dendritic cells, $10^6$ UCBMC were stimulated for 16 hr at 37 °C in RPMI 1640 medium supplemented with 10% FBS in the presence or absence of 1 µg/mL LPS (TLR4 ligand, *E. coli* 055:B5; Invivogen, San Diego CA). Brefeldin A (Sigma, St. Louis MO) was added after 1 hr incubation. Cells were stained for APC-Cy7-CD14 and PCP-Cy5.5-HLA-DR, fixed, permeabilized, and stained intracellularly for APC-TNFα and PE-IL-6. All samples were acquired with the Attune NxT Flow Cytometer (ThermoFisher Scientific, Waltham, MA) and analyzed using FlowJo 10.5.0 (Ashland, OR).

## Pathogen stimulation

Approximately $5x10^5$ MACS purified UCB monocytes were cultured with RSV (Human respiratory syncytial virus ATCC VR-1540, Manassas, VA) or *E. coli* (*Escherichia coli* (Migula) Castellani and Chalmers ATCC 11775, Manassas, VA) or left untreated in RP10 medium for 16 hr at 37 °C. RSV was added at a multiplicity of infection (MOI) of 5 and *E. coli* at $6x10^5$ cfu/mL. Following the 16 hr incubation, cells were spun down. Cell pellets were frozen in QIAzol lysis reagent (Qiagen, Hilden Germany) to generate RNA-Seq libraries. Cell supernatants were frozen at –80°C to measure the concentrations of chemokines and cytokines using Luminex.

## Luminex and ELISA

Immune mediators in plasma were measured using a customized multiplex human factor panel (R & D Systems, Minneapolis MN) measuring cytokines (IFNβ, IFNγ, IL-1β, IL-10, IL-12p70, IL-13, IL-15, IL-17A, IL-18, IL-1RA, IL-2, IL-21, IL-4, IL-5, IL-7, TNFα, IL-23, IL-31, IL-22, IL-27), chemokines (CCL2/MCP-1, CCL3/MIP-1α, CCL4/MIP-1β, CCL5/RANTES, CCL11/Eotaxin, CXCL1/GROα, CXCL8/IL-8, CXCL9/MIG, CXCL10/IP-10, CXCL11/I-TAC, CXCL12/SDF-1α, CXCL13/BCA-1), growth factors (BDNF, GM-CSF, HGF, EGF, VEGF, PDGF-BB) and additional molecules (PD-L1, S100). Metabolic hormones were measured using a 3-plex kit measuring insulin, leptin, and PYY (Millipore, Burlington MA). Adipokines were assayed using a 5-plex kit measuring adiponectin, adipsin, lipocalin-2, total PAI-1, and resistin (Millipore, Burlington MA). CRP and IL-6 were measured in UCB plasma using a high-sensitivity ELISA (Life Technologies, Carlsbad CA) per the manufacturer's instructions.

Supernatants from fetal rhesus macaque monocyte stimulation experiments were analyzed using an NHP XL Cytokine Premixed 36-plex kit (Bio-Techne, Minneapolis MN). Samples were diluted per the manufacturer's instructions and analyzed in duplicate on the Magpix Instrument (Luminex, Austin, TX). Data were fit using a 5P-logistic regression on xPONENT software.

## Bulk RNA-Seq library generation

Total RNA was isolated from monocytes using an mRNeasy kit (Qiagen, Valencia CA). Quality and concentrations were measured using Agilent 2100 Bioanalyzer. Libraries were generated using the TruSeq Stranded Total RNA-Seq kit (Illumina, San Diego CA). Briefly, following rRNA depletion, mRNA was fragmented for 8 min, converted to double-stranded cDNA, and adapter ligated. Fragments were then enriched by PCR amplification and purified. The size and quality of the library were verified using Qubit and Bioanalyzer. Libraries were multiplexed and sequenced on the HiSeq4000 platform (Illumina, San Diego CA) to yield an average of 20 million 100 bp single-end reads per sample.

## Bulk RNA-Seq analysis

Quality control of raw reads was performed using FASTQC retaining bases with quality scores of ≥20 and reads ≥35 base pairs long. Reads were aligned to the human genome (hg38) using splice-aware aligner TopHat using annotations available from ENSEMBL (GRCh38.85) database. Lowly expressed genes were filtered at the counting stage, eliminating genes with 0 counts in at least three samples regardless of the group. Quantification of read counts was performed using the GenomicRanges package in R and normalized to derive transcripts per million (TPM) counts.

Responses to RSV and *E. coli* were modeled pairwise relative to unstimulated samples using negative binomial GLMs following low read count filtering. Genes with $log_2$ fold change ≥1 in either

direction and corrected expression difference FDR <0.05 were considered differentially expressed genes (DEG). Functional enrichment of DEG was performed using Metascape (*Zhou et al., 2019*). Heatmaps of fold change or TPMs and bubble plots of enrichment of Gene Ontology (GO) terms were generated using ggplot in R.

## Cell Sorting and library generation for single cell (sc)RNA-seq

UCBMC were thawed then washed twice in PBS with 0.04% BSA and incubated with individual 3′ CellPlex oligos (10 X Genomics) per manufacturer's instructions for 5 min at room temperature. Pellets were washed three times in PBS with 1% BSA, resuspended in 300 µL FACS buffer, and sorted on BD FACS Aria Fusion into RPMI (supplemented with 30% FBS) following the addition of SYTOX Blue stain (1:1000, ThermoFisher) for live versus dead discrimination. Sorted cells were counted in triplicates and resuspended in PBS with 0.04% BSA in a final concentration of 1200 cells/µL. Cells were immediately loaded in the 10 X Genomics Chromium with a loading target of 17,600 cells. Libraries were generated using the V3.1 chemistry (gene expression) and Single Cell 3′ Feature Barcode Library Kit per the manufacturer's instructions (10 X Genomics, Pleasanton CA). Libraries were sequenced on Illumina NovaSeq 6000 with a sequencing target of 50,000 gene expression reads and 5000 multiplexed CellPlex reads per cell.

## scRNA-seq data analysis

For 3′ gene expression with CellPlex, raw reads were aligned and quantified using Cell Ranger (version 6.0.2, 10 X Genomics) against the human reference genome (GRCh38) using the *multi* option and CMO information. Only singlets identified from each sample were included in downstream analyses. Droplets with ambient RNA (cells fewer than 400 detected genes), potential doublets (cells with more than 4000 detected genes) and dying cells (cells with more than 20% total mitochondrial gene expression) were excluded during initial QC. Data objects from lean and obese groups were integrated using Seurat. Data normalization and variance stabilization was performed using the *SCTransform* function using a regularized negative binomial regression, correcting for differential effects of mitochondrial and ribosomal gene expression levels and cell cycle. Dimension reduction was performed using the *RunPCA* function to obtain the first 30 principal components followed by clustering using the *FindClusters* function in Seurat. Visualization of clusters was performed using the UMAP algorithm as implemented by Seurat's *runUMAP* function. Cell types were assigned to individual clusters using the *FindMarkers* function with a fold change cutoff of at least 0.5 and using a known catalog of well-characterized scRNA markers for PBMC (*Zheng et al., 2017*).

Differential expression analysis was tested using the Wilcoxon rank sum test followed by Bonferroni correction using all genes in the dataset. For gene scoring analysis, we compared gene signatures and pathways from KEGG (https://www.genome.jp/kegg/pathway.html) in subpopulations using Seurat's *AddModuleScore* function. All graphs were generated in R.

## Phagocytosis assay

To quantify the phagocytic ability of UCB monocytes, monocytes were isolated using magnetically activated cell sorting and anti-CD14 antibody coupled to magnetic beads per the manufacturer's recommendation (MACS, Miltenyi Biotech, San Jose, CA). Cells were activated with 1 µg/ml LPS for 16 hr, then cultured with *E. coli* particles conjugated with HRP (horseradish peroxidase) in 96-well plates per manufacturer's protocol (CytoSelect 96-well phagocytosis assay, Cell Biolabs, San Diego CA) for 3 hr in 37 °C incubator with 5% $CO_2$. Cells were washed, fixed, permeabilized, incubated with substrate, and quantified using colorimetry (CytoSelect 96-well phagocytosis assay, Cell Biolabs, San Diego CA).

## Cell migration assay

The migratory potential of monocytes to supernatants containing chemokines was measured using the CytoSelect 96-well Cell Migration Assay Cell Migration assay (Cell Biolabs, San Diego CA). Briefly, $2 \times 10^5$ purified UCB monocytes isolated using anti-CD14 antibody coupled to magnetic beads per the manufacturer's recommendation (MACS, Miltenyi Biotech, San Jose, CA) were incubated in serum-free media in the upper wells of the migration plate, while supernatants collected following PMA (phorbol myristate acetate) stimulation of adult PBMC were placed in lower wells and incubated at

37 °C and 5% $CO_2$ for 5 hr. The number of cells that migrated into the lower wells was quantified using CyQuant cell proliferation assay per the manufacturer's instructions. Absolute numbers of migrated cells were calculated using a standard curve for CyQuant assay with a linear range of fluorescence limited from 50 to 50,000 cells. Cell-free media served as a negative control.

## Bulk ATAC-Seq

ATAC-Seq libraries were generated using OMNI-ATAC to reduce mitochondrial reads (*Araujo et al., 2011*). Briefly, 50,000 purified UCB monocytes were stimulated with 1 µg/mL LPS for 16 hr before being lysed in lysis buffer (10 mM Tris-HCl (pH 7.4), 10 mM NaCl, 3 mM $MgCl_2$), for 3 min on ice to prepare the nuclei. Immediately after lysis, nuclei were spun at 500 g for 10 min to remove the supernatant. Nuclei were then incubated with a transposition mixture (100 nM Tn5 transposase, 0.1% Tween-20, 0.01% Digitonin, and TD Buffer) at 37 °C for 30 min. Transposed DNA was then purified with AMPure XP beads (Beckman Coulter) and partially amplified for five cycles using the following PCR conditions - 72 °C for 3 min; 98 °C for 30 s and thermocycling at 98 °C for 10 s, 63 °C for 30 s, and 72 °C for 1 min. To avoid overamplification, qPCR was performed on 5 µL of partially amplified library. Additional cycles of amplification for the remainder of the sample were calculated from the saturation curves (cycles corresponding to a third of the saturation value). Fully amplified samples were purified with AMPure beads and quantified on the Bioanalyzer (Agilent Technologies, Santa Clara CA). Libraries were sequenced on the HiSeq4000 platform (Illumina, San Diego CA).

## Analysis of bulk ATAC-Seq data

Paired reads from sequencing were quality-checked using FASTQC and trimmed to retain reads with quality scores of ≥20 and minimum read lengths of 50 bp. Trimmed paired reads were aligned to the human genome (hg38) using Bowtie2 (–X 2000 –k 1 –very-sensitive –no-discordant –no-mixed). Reads aligning to mitochondrial genome and allosomes were removed using samtools. PCR duplicate artifacts were then removed using Picard. Finally, poor quality alignments and improperly mapped reads were filtered using samtools (samtools view –q 10 –F 3844). To reflect the accurate read start site due to Tn5 insertion, BAM files were repositioned using the ATACseqQC package in R. The positive and negative strands were offset by +4 bp and –5 bp, respectively. Samples within a group were merged and sorted using samtools.

Sample QC and statistics for merged BAM files were generated using HOMER makeTagDirectory function. Transcription Start Site (TSS enrichment) for each sample was assessed using ChIP-Seeker. Accessible chromatin peaks were called for mapped paired reads using the HOMER findpeak function (-minDist 150 –region –fdr 0.05). Differentially accessible regions (DAR) in either direction were captured using HOMER getDiffererentialPeaks function (-q 0.05). DAR were annotated using the human GTF annotation file (GRCh38.85) and ChIPSeeker with a promoter definition of –1000 bp and +100 bp around the transcriptional start site (TSS). Peaks overlapping 5'UTRs, promoters, first exons, and first introns were pooled for functional enrichment of genes. For intergenic changes, the genes closest to the intergenic DAR were considered. Functional enrichment of this pooled list of genes was performed using DAVID (Fisher p-value <0.05). BAM files were converted to bigWig files using bedtools and visualized on the new WashU EpiGenome browser. Finally, motif analyses of over-representative transcription factor binding sites for differential regions were analyzed using HOMER's *FindMotifsGenome* function with -size parameter set at 75, optimized for transcription factors.

## scATAC sample preparation

One to two million UCBMC (n=4/group) were incubated for 4 hr at 37 °C in the presence or absence of 1 µg/mL LPS. Cell pellets were then washed, and surface-stained with monocyte markers CD14-AF700 and HLA-DR APC-Cy7 for 30 min. Stained cells were washed, resuspended in FACS buffer, and stained for live-dead exclusion (SYTOX Blue, 1:1000 dilution). Equal numbers of monocytes (CD14 +HLA-DR+) were sorted and pooled by group (lean/obese) into RPMI supplemented with 30% FBS. Cells were washed thoroughly, and nuclei were isolated using the low cell input nuclei isolation protocol (10 X Genomics). Nuclei were counted and verified for integrity then resuspended in PBS with 0.04% BSA at 1000 nuclei/µL concentration. Monocyte nuclei were loaded onto the 10 X Genomics Chromium according to the manufacturer's protocol using the single-cell ATAC kit (v2). Library preparation was performed per the manufacturer's protocol prior to sequencing on Illumina NovaSeq 6000 platform.

## scATAC data analysis

Basecall files were used to generate FASTQ files with cellranger-atac (v1; 10 X Genomics). Reads were aligned to the human genome using cellranger-atac count with the cellranger-atac-GRCh38.1.1.0 reference. Mapped Tn5 insertion sites from cellranger were read into ArchR (version 1.0.1) R package retaining barcodes with at least 1000 fragments per cell and a TSS enrichment score >4. Doublets were identified and filtered using *addDoubletScores* and *filterDoublets* (filter ratio = 1.4) respectively before iterative LSI dimensionality reduction (iterations = 2, res = 0.2, variable features = 25000, dim = 30). Clustering was then performed (*addClusters*, res = 0.8) before UMAP dimensionality reduction (nNeighbors = 30, metric = cosine, minDist = 0.4). One cluster enriched for high doublet scores and was removed. Peaks for each cluster were calculated using MACS2, using the *addReproduciblePeakSet* function. Marker peaks for each cluster and differential peaks with stimulation were calculated using the *getMarkerFeatures* function using the Wilcoxon test. A cluster of activated monocytes was identified by pileups and feature plots of canonical cytokine and activation markers.

## Cut&Tag

Chromatin immunoprecipitation sequencing (ChIP-seq) was performed using the Cleavage Under Targets and Tagmentation (CUT&Tag) approach using roughly 150–200 K purified resting monocytes (n=3/group). Further processing, chromatin extraction, fragmentation, antibody precipitation, and library preparation for H3K9me3 (Cat # 39765) were performed at Active Motif (Carlsbad, CA). Libraries were sequenced on the Illumina NextSeq550. Paired ended reads (38 bp) were aligned to human genome hg38 using BWA with default settings (bwa mem). Peaks were called using the MACS algorithm at a cutoff of p-vaue 1e-7, without control file. Due to the low numbers of peaks in several samples, aligned BAM files were merged by group (n=3/group). Peaks were determined using HOMER's *findPeaks* function. Differential analysis between the two groups was performed using HOMER's *getDifferentialPeaks* function. Peaks were annotated using ChIPseeker.

## Metabolic assays

Oxygen Consumption Rate (OCR) and Extracellular Acidification Rate (ECAR) were measured using Seahorse XF Glycolysis Rate Assay on Seahorse XFp Flux Analyzer (Agilent Technologies, Santa Clara, CA) following the manufacturer's instructions. Briefly, 200,000 purified monocytes (pooled n=3/group) were seeded in glucose-free media and cultured in the presence/absence of 1 µg/mL LPS for 1 hr in a 37 °C incubator without $CO_2$ on Cell-Tak (Corning, Corning, NY) coated eight-well culture plates in phenol-free RPMI media containing 2 mM L-glutamine, 1 mM sodium pyruvate, and 5 mM HEPES. Plates were run on the XFp for 8 cycles of basal measurements, followed by acute injection of L-glucose (100 mM), oligomycin (50 uM), and 2-DG (500 mM). Data were analyzed on Seahorse Wave desktop software (Agilent Technologies, Santa Clara, CA).

## Histone ELISA

Nuclear extracts from $2 \times 10^5$ UCB monocytes purified using an anti-CD14 antibody coupled to magnetic beads per the manufacturer's recommendation (MACS, Miltenyi Biotech, San Jose, CA) were isolated per the manufacturer's instructions (Abcam, Cambridge UK) and quantified using a micro-BCA assay protein assay kit (ThermoFisher Scientific, Waltham, MA). Histone modifications were measured using a Histone H3 modification Multiplex Assay (Abcam, Cambridge UK). The input was normalized based on total protein concentration, and 20 ng of nuclear extract was added to each well. Given limited sample availability, only a subset of histone methylation marks were assayed (H3K4me1, H3K4me2, H3K4me3, H3K9me3, H3K9Ac, and total H3). Optical density was measured at 450 nm. All values are reported as percentages of the total H3 signal.

## Histone flow

Activation-induced changes in histone post-translational modifications were assayed using flow cytometry (n=8/group). Briefly, $10^6$ UCBMC were stimulated with 1 µg/mL LPS or left untreated for 2 hr in a 37 °C incubator with 5% CO2. Cell pellets were washed with FACS buffer, surface stained for monocytes (CD14 AF700, HLA-DR APC-Cy7) for 20 min, washed, and fixed using Foxp3/Transcription factor Fix/Perm buffer (Tonbo Biosciences, San Diego, CA) for 1 hr at 4 °C. Pellets were then stained intracellularly overnight with antibodies against H3K4me3-AF647 (Clone EPR20551-225,

Abcam, Cambridge UK) and H3K9me3-PE (Clone D4W1U, Cell Signaling Technology, Danvers, MA). Cells were washed twice with Perm buffer (Tonbo Biosciences, San Diego, CA), followed by FACS buffer. After the final wash, pellets were resuspended in FACS buffer and analyzed on Attune NxT flow cytometer (ThermoFisher Scientific, Waltham, MA).

## Phospho-Flow

Activation-induced changes in signaling mediators downstream of TLR4 were measured using flow cytometry. Briefly, $10^6$ UCBMC (n=8/group) were stimulated either for 30 min or 2 hr with 1 µg/mL LPS or left untreated in a 37 °C incubator with 5% CO2. Cells were washed with FACS buffer and surface stained for CD14 and HLA-DR in FACS tubes. Pellets were washed in FACS buffer and resuspended in 100 µL prewarmed PBS (Ca + Mg + free). Cells were fixed immediately by the addition of equal volumes of prewarmed Cytofix Buffer (BD Biosciences, Brea, CA) and thorough mixing and incubating at 37 °C for 10 min. Cells were then centrifuged at 600 g for 8 min. Supernatants were removed leaving no more than 50 µL residual volume. Cells were then permeabilized by the addition of 1 mL 1 X BD PermWash Buffer I, mixed well, and incubated at room temperature for 30 min. Pellets were then washed and stained intranuclearly with antibodies against NF-kB p65 pS529 AF647 (Clone K10-895.12.50, Cell Signaling Technology, Danvers, MA) (for 30 min stimulation samples) or IkBa PE (Clone MFRDTRK, eBioscience, San Diego CA) and Phospho-p38 MAPK-APC (Clone 4NIT4KK, eBioscience, San Diego CA) for 1 hr at room temperature in the dark. Samples were washed twice in Permwash Buffer I, resuspended in FACS buffer, and analyzed on Attune NxT flow cytometer.

## Imaging flow cytometry

Nuclear translocation of p50 was measured using the Amnis NFkB Translocation kit (Luminex Corporation, Austin TX). Briefly, 1–2 x $10^6$ UCBMC were stimulated with 1 µg/mL LPS for 4 hr in a 37 °C incubator with 5% $CO_2$. Cell pellets were washed thoroughly and stained for dead cell exclusion (Ghost Violet 510, 1:1000 dilution Tonbo Biosciences, San Diego, CA) for 30 min at 4 °C. Cells were then washed, and surface stained for 20 min at 4 °C (CD14-APC, HLA-DR-APC-Cy7). Cells were washed in 1 X Assay Buffer (Luminex Corporation, Austin TX) and fixed using 1 X Fixation buffer for 10 min at room temperature. Washed pellets were then stained intranuclearly for p50-AF488 (1:20 dilution) in 1 X Assay Buffer for 30 min at room temperature in the dark. At the end of the incubation, cells were washed twice in 1 X Assay buffer and resuspended in 50 µL 0.25 X Fixation Buffer in polypropylene Eppendorf tubes. Samples were analyzed on the Amnis Imagestream XMark II imaging flow cytometry platform and analyzed on the IDEAS Software using the nuclear translocation wizard.

## Fetal rhesus macaque samples macrophage isolation and stimulation

PBMC, ileal lamina propria lymphocytes (LPL), and splenocytes from gestational day (GD) 130 rhesus macaque fetuses born to obese and lean dams (n=3/group) were isolated and cryopreserved from animals described in *Sureshchandra et al., 2022*. PBMC were stimulated with 1 µg/mL LPS for 16 hr in a 37 °C incubator with 5% CO2. The frequency of responding monocytes was determined using intracellular cytokine staining for IL-6 and TNFα following surface staining for CD14 + HLA-DR + cells. Ileal LPLs and splenocytes were stained using a antibodies directed against: CD14, HLA-DR, CD206, CD209, CD169, CD163, and CD64. Additionally, 10,000 CD14$^{high}$HLA-DR+ splenic macrophages and 5,000 ileal CD14$^{high}$HLA-DR$^+$ macrophages from each sample were plated and subsequently stimulated with *E. coli* (6x$10^5$ cfu/mL) or left untreated for 16 hr at 37 °C. Plates were spun and supernatants were collected for analysis of secreted cytokines and chemokines using Luminex.

## Statistical analysis

All statistical analyses were conducted in Prism 8 (GraphPad, San Diego CA). For cord blood cytokine, immune phenotyping, and ex vivo responses readouts, data was tested for normality using the Shapiro-Wilk test (alpha = 0.05), analysis of equal variance (F-test) and all outliers in two-way and four-way comparisons identified using ROUT analysis (Q=0.1%). If data were normally distributed across all groups, differences with obesity and pregnancy were tested using ordinary one-way ANOVA with unmatched samples. Multiple comparisons were corrected using the Holm-Sidak test adjusting the family-wise significance and confidence level at 0.05. If the Gaussian assumption was not satisfied, differences were tested using the Kruskal-Wallis test (alpha = 0.05) followed by Dunn's multiple

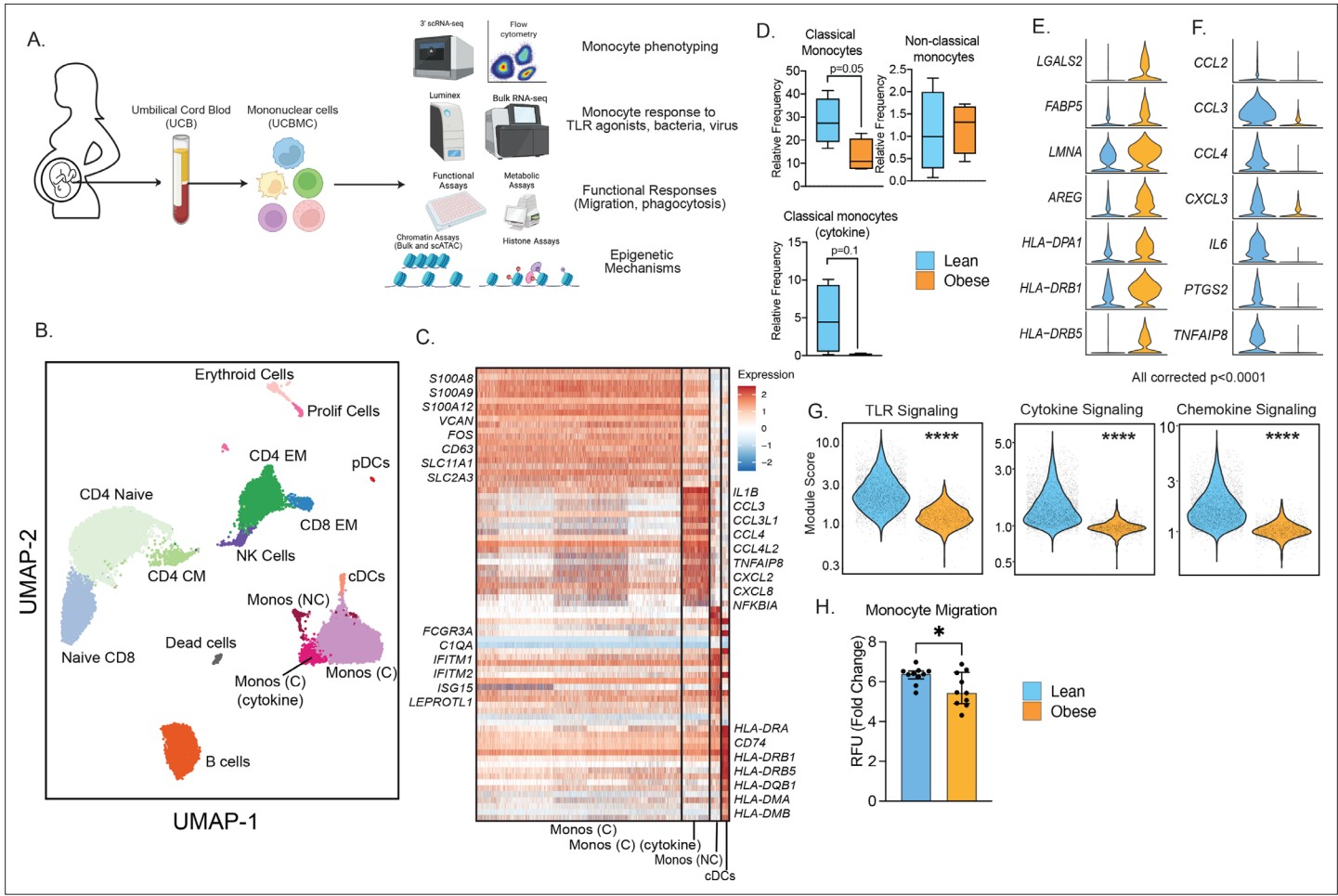

**Figure 1.** Experimental design and phenotypic changes in UCB monocytes. (**A**) Experimental design for the study. Cord blood samples were obtained from neonates born to lean and mothers with obesity (n=43 lean and n=36 obese, *Table 1*). UCBMC and plasma were isolated and used to assess the impact of pregravid obesity on fetal immunity using genomic (bulk and single-cell RNA and ATAC-Seq, ChIP-Seq) and functional assays (flow cytometry, ex vivo stimulations, phagocytosis, and migration). (**B**) UMAP of term cord blood mononuclear cells collected from lean mothers and mothers with obesity (n=4/group). Samples were hashed using CellPlex (10 X Genomics), sorted for live cells, and analyzed using a 10 X single-cell 3' gene expression assay. (**C**) Heatmap of top 30 markers of monocyte and conventional DC clusters in UCBMC. (**D**) Box and whisker plots comparing relative frequencies of monocyte subsets (mean and ± SEM). (**E–F**) Violin plots of the top differentially expressed genes (FDR p<0.0001) in cord blood classical monocytes – (**E**) upregulated and (**F**) downregulated with maternal obesity. (**G**) Violin plots comparing module scores of TLR, cytokine, and chemokine signaling in cord blood classical monocytes with maternal obesity. (**H**) Bar graph comparing migration potential of UCB monocytes in response to supernatants from PMA-stimulated adult PBMC. Fold changes were calculated relative to no stimulation controls (mean and± SEM). * - p<0.05.

The online version of this article includes the following figure supplement(s) for figure 1:

**Figure supplement 1.** Longitudinal changes in circulating inflammatory environment.

**Figure supplement 2.** Innate immune cell responses in cord blood.

**Figure supplement 3.** Single cell profiling of umbilical cord blood mononuclear cells.

hypothesis correction tests. Differences in normally distributed datasets were tested using an unpaired t-test with Welch's correction (assuming different standard deviations). Two group comparisons that failed normality tests were carried out using the Mann-Whitney test. All bar graphs report mean and ± standard error of the mean (SEM).

## Results

### Cord blood monocytes from babies born to mothers with obesity exhibit attenuated responses to TLR stimulation

To dissect the mechanisms by which pregravid obesity impacts functional responses in cord blood monocytes, we carried out a multi-pronged systems immunology approach (*Figure 1A*). We began by characterizing immune cell composition and phenotypes in cord blood of subjects stratified by their pre-pregnancy body mass index (BMI) – babies born to lean mothers (BMI <25) and babies born to mothers with obesity (BMI >30). As previously reported, maternal obesity did not alter the frequencies of white blood cells (WBC) or their subsets in the cord blood (*Wilson et al., 2015*; *Sureshchandra et al., 2021c*; *Sureshchandra et al., 2017*; *Wilson and Messaoudi, 2015*; *Figure 1—figure supplement 1A*). Additionally, no changes were observed in cord blood plasma levels of inflammatory markers (IL-6, CRP) or insulin (*Figure 1—figure supplement 1B–C*). Leptin levels were, however, significantly higher in the obese group (*Figure 1—figure supplement 1C*).

Multi-parameter flow cytometry analysis of innate immune cells within umbilical cord blood mononuclear cells (UCBMC, *Figure 1—figure supplement 2A*) revealed no differences in dendritic cell frequencies (*Figure 1—figure supplement 2B*) but a significant drop in plasmacytoid DCs (pDCs) in the obese group (*Figure 1—figure supplement 2C*). Additionally, proportion of total monocytes (CD14 + HLA-DR+) within live mononuclear cells was significantly lower in the obese group (*Figure 1—figure supplement 2D*). However, monocyte subsets - classical, non-classical, and intermediate remained comparable between the two groups (*Figure 1—figure supplement 2D*). Finally, UCB monocytes from the obese group responded poorly to ex vivo LPS (TLR4 agonist) stimulation (*Figure 1—figure supplement 2F*), recapitulating the phenotype observed in previously reported studies (*Wilson et al., 2015*; *Sureshchandra et al., 2021c*; *Sureshchandra et al., 2017*; *Wilson and Messaoudi, 2015*). These differences were not driven by differences in the mode of delivery or the viability of cells between the two groups (*Figure 1—figure supplement 2G–H*).

### Single-cell RNA sequencing reveals maternal obesity-associated transcriptional shifts in the UCB monocytes

We next asked if cord blood monocyte subsets were transcriptionally primed to respond poorly to ex vivo stimulation signals. To test this, we performed droplet-based single-cell RNA sequencing (scRNA-Seq) of UCBMC from lean and obese groups (n=4/group, hashed using Cell Plex). Integration of all donors using Recursive Principal Component Analysis (rPCA) and Uniform Manifold Approximation and Projection (UMAP) analysis revealed the expected major immune subsets (*Figure 1B*) including T cells (*CD3D*, *CD8A*), comprised mostly of naïve cells (*CCR7*, *IL7R*) and some memory T cells lacking *CCR7*; B cells (*CD79A*, *MS4A1*); NK cells (*NCAM1*); conventional dendritic cells (cDCs, *HLA-DRA*, *FCER1A*); plasmacytoid dendritic cells (pDCs, *IRF8*); monocytes (*CD14*); erythroid cells (*HBB*); and a small subset of proliferating cells (*MKI67*) (*Figure 1—figure supplement 3A*). Assessment of individual group (*Figure 1—figure supplement 3B*) and donor contributions (*Figure 1—figure supplement 3C* and *Supplementary file 1*) highlighted representation of every sample (lean and obese) in individual clusters identified, with modest increases in proportions of naïve CD4 and CD8 T cell in obese group (*Figure 1—figure supplement 3D*).

A closer look at the monocytes revealed the presence of 3 clusters (*Figure 1B*), all expressing *CD14*: classical monocytes (Mono C) expressing higher *S100A8*, *S100A9*, and *VCAN*; non-classical monocytes (Mono NC) expressing higher levels of *FCGR3A* and interferon-stimulated genes; and a cluster of classical monocytes (Mono (C) cytokine) with elevated levels of inflammatory transcripts *IL1B*, *CCL3*, *CXCL2*, and *CXCL8* (*Figure 1C*). Comparisons of the relative abundance of these subsets revealed a decrease in Mono C (p=0.05) and a non-significant decrease in Mono (C) cytokine clusters (p=0.1) with pregravid obesity (*Figure 1D* and *Figure 1—figure supplement 3D*). This shift in UCB monocyte cell states is further demonstrated by differential gene expression analysis (*Figure 1E and F*) and module score comparisons (*Figure 1G* and *Figure 1—figure supplement 3E*) within UCB monocyte clusters. Specifically, pregravid obesity resulted in elevated expression of genes important for antigen presentation and immune regulation (*Figure 1E*). On the other hand, immune signatures associated with cytokine (*IL6*), chemokine (*CCL2*, *CCL3*, *CXCL3*), and TLR signaling were attenuated in the obese group (*Figure 1F and G*), in line with reduced TLR responses reported here

(*Figure 1—figure supplement 2F*) and prior studies (*Sureshchandra et al., 2017*). Finally, dampened chemokine signaling translated to a reduction in the migration capacity of purified monocytes (*Figure 1H*).

## Pregravid obesity compromises cord blood monocyte responses to bacteria

Given the increased incidence of bacterial and viral infections in babies born to mothers with obesity (*Griffiths et al., 2016*; *Rastogi et al., 2015*; *Suk et al., 2016*), we next interrogated if the reduced UCB monocyte response to LPS stimulation (*Wilson et al., 2015*; *Sureshchandra et al., 2021c*; *Sureshchandra et al., 2017*; *Wilson and Messaoudi, 2015*) extended to anti-microbial response. To that end, monocytes were purified from UCBMC and cultured with *E. coli* for 16 hr at 37 °C. Secreted cytokines and chemokines were profiled using Luminex, while the transcriptional response to infection was recorded using bulk RNA sequencing (*Figure 2A*). While comparable levels of TNFα were secreted by both groups in absence of stimulus, an infection-induced increase in the secretion of pro-inflammatory cytokines (IL-6, IL12p70), regulatory cytokines (IL-10, PD-L1), chemokines (CCL4, CCL11, CXCL11), and growth factors (GM-CSF) was only detected in the lean group after *E. coli* stimulation (*Figure 2B*).

Next, we examined differences in transcriptional responses to *E. coli*. The number of differentially expressed genes (DEG) relative to no stimulation was higher in the lean group (*Figure 2—figure supplement 1A*). While ~30% of genes up-regulated with infection (160 genes) were shared between the lean and obese groups (*Figure 2C*), the fold change of several of these genes involved in myeloid cell activation (*CSF2, CSF3, CCR7*) and cytokine signaling (*IL6, IL23A, IL10, IL12B, CCL2, CCL3, CCL4, CCL8, CCL20*) was lower in the obese group (*Figure 2D*). Importantly, functional enrichment of genes upregulated in the lean group alone revealed an over-representation of pathways associated with immune regulation (positive regulation of protein phosphorylation and regulation of cellular response to stress) (*Figure 2C*). DEG enriching to these pathways include cytokines (*IL1B, IL1RN*), chemokines (*CXCL1, CXCL3*), growth factors (*VEGFA, FGF2*), and signaling molecules (*CASP5, NFKBIZ, SOCS3, TLR8, MMP9*) indicating a robust innate immune response in the lean group (*Figure 2E*). We also observed ~50% overlap in genes downregulated with infection (*Figure 2—figure supplement 1B*). These gene signatures enriched to GO terms associated with myeloid cell activation and negative regulation of differentiation (*Figure 2—figure supplement 1C*). Interestingly, downregulation of MHC class II molecules (*HLA-DPA1, DPB1, DQB1, DRA, DRB1, DRB5*) was observed exclusively in the lean group (*Figure 1—figure supplement 2C and D*).

Given the modest inflammatory response to *E. coli* infection by UCB monocytes from the obese group, we next compared their phagocytic ability (*Figure 2F*). Purified UCB monocytes were cultured with labeled *E. coli* and probed for internalization. UCB monocytes from the obese group were more phagocytic compared to the lean group (*Figure 2G*), suggesting a more regulatory phenotype.

## Attenuated antiviral transcriptional responses in UCB monocytes with pregravid obesity

Previous studies in rodent models have reported dysregulated inflammatory responses in the lungs of pups born to obese dams following RSV infection (*Griffiths et al., 2016*). Given that RSV is sensed by myeloid cells via TLR4/TLR8 pathways, we asked if maternal obesity compromises fetal myeloid cell responses to ex vivo RSV infection (*Figures 2A and 3*). Interestingly, we saw no differences between the lean and obese groups in terms of immune mediator production 16 hr post-stimulation (*Figure 3A*). UCB monocytes from the obese group generated a larger transcriptional response than those from the lean group (*Figure 3—figure supplement 1A*). While gene signatures associated with a robust response to the virus were observed in both groups (*Figure 3B and C*), significant differences in DEGs upregulated with RSV stimulation were noted (*Figure 3B*). Particularly, DEGs unique to the lean group were associated with anti-viral effector responses (*Figure 3C*). This list included genes involved in TLR signaling (*MYD88, GSDMD, CASP1*), cytokine and chemokine signaling (*CCL2, IL1RN, IL10*), and effector molecules that initiate a robust Th1 response (*STAT4, CXCL10, IL15, IL27*) (*Figure 3C and D*).

Moreover, following RSV infection, several interferon-associated genes (ISGs) (*STAT4, IFNA1, IRF1,* and *IRF5*) were upregulated exclusively in the lean group (*Figure 3D*) while other ISGs were poorly induced in the obese group (*Figure 3—figure supplement 1B*). Given comparable levels of secreted

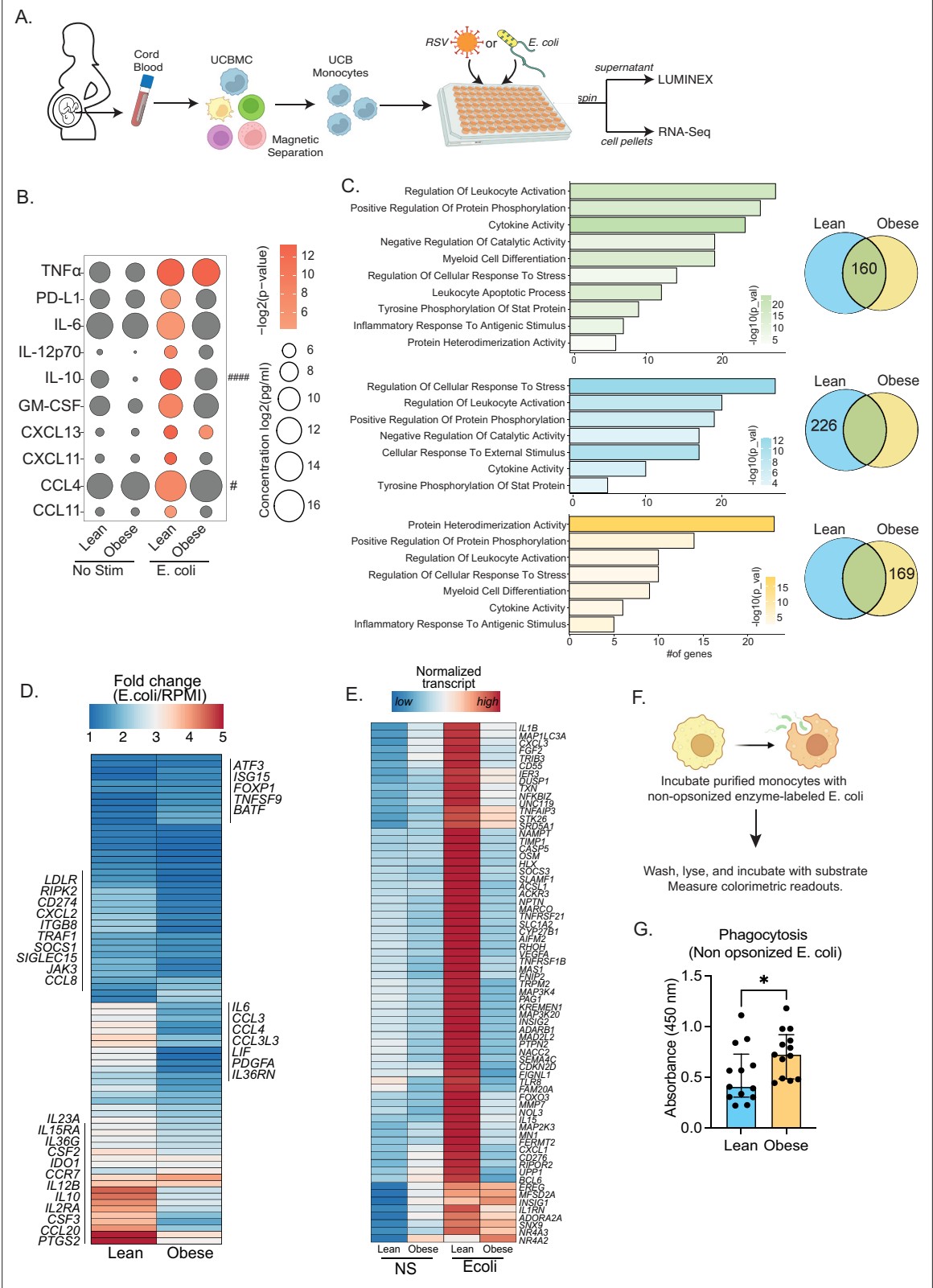

**Figure 2.** Cord blood monocyte responses to ex vivo *E. coli* infection are attenuated with maternal obesity. (**A**) Experimental design for in vitro bacterial and viral stimulation. Purified monocytes were cultured in the presence/absence of *E. coli* or RSV for 16 hr. Cell pellets were used for bulk RNA-Seq analyses and supernatants were used for Luminex analyses of secreted cytokines and chemokines. (**B**) Bubble plot of key secreted factors significantly different following *E. coli* infection. The size of the bubble represents the quantity of the secreted analyte (log-transformed) whereas color

*Figure 2 continued on next page*

*Figure 2 continued*

represents statistical significance relative to no stimulation controls. Statistically significant analytes between lean and obese groups are highlighted with # - p<0.05. (**C**) Venn diagram (right) and corresponding functional enrichment (left) of genes upregulated with *E. coli* infection in lean and obese groups (Green denotes common DEG, blue DEG unique to lean group, and yellow DEG unique to the obese group) using metascape. The length of the bar indicates the number of genes in each gene ontology (GO) term. Color represents the statistical significance of each GO term. (**D**) Heatmap comparing fold changes of the genes upregulated in both groups (77 genes) that mapped to GO terms 'myeloid cell differentiation', 'inflammatory response to antigenic stimulus', 'leukocyte apoptotic process', 'regulation of leukocyte activity', 'cytokine activity', and 'positive regulation of protein phosphorylation'. (**E**) Heatmap comparing normalized transcript counts (blue – low to red – high) of genes exclusively upregulated in the lean group following *E. coli* infection. (**F**) Experimental design for measuring ex vivo phagocytosis by cord blood monocytes. (**G**) Bar graph depicting colorimetric readout of phagocytosed *E. coli* particles by UCB monocyte. * or /#- p<0.05, #### - p<0.0001.

The online version of this article includes the following figure supplement(s) for figure 2:

**Figure supplement 1.** Cord blood monocyte responses to *E. coli*.

type-I IFN following 16 hr RSV infection (*Figure 3A*), these data suggested potential defects in the response to type I IFN in UCB monocytes from the obese group. To test this hypothesis, we stimulated total UCBMC with a cocktail of IFNα and IFNβ for 6 hr and measured activation markers using flow cytometry (*Figure 3E*). The frequency of cells expressing IFN receptor (IFNAR1) was comparable between the groups and treatment conditions as was the increase in mean fluorescence intensity (MFI) following stimulation (*Figure 3—figure supplement 1C*). Nevertheless, activation markers CCR7, CD40, and CD86 were upregulated to a lesser extent in the obese group (*Figure 3F*). Finally, analysis of DEG downregulated after RSV stimulation (*Figure 3—figure supplement 1D*) in the lean group showed enrichment to transcription and translation while DEG unique to the obese group enriched to fatty acid and immunoglobulin binding (*Figure 3—figure supplement 1E*).

Finally, a comparison of overall transcriptional responses to *E. coli* and RSV revealed limited overlap in DEG, an expected outcome given the different nature of the pathogens (*Figure 3—figure supplement 1F*). Interestingly, the inflammatory response to RSV was less pronounced compared to that generated in response to *E. coli* (*Figure 3—figure supplement 1G*). However, overall responses to both *E. coli* and RSV were dampened with maternal obesity.

## UCB monocytes from babies born to mothers with obesity are epigenetically and metabolically poised to respond poorly to LPS

Prior studies from our laboratory have shown that maternal obesity is associated with loss of promoter methylation of several negative regulators of monocyte activation such as PPARγ (*Sureshchandra et al., 2017*; *Sureshchandra et al., 2019*) in UCB monocytes. Given the prominent role of chromatin accessibility changes during acute responses to pathogens, we asked if UCB monocytes from babies born to mothers with obesity are epigenetically poised for dysfunctional responses to TLR ligands. To test this, we isolated nuclei from purified resting CD14 +monocytes and identified global differences in baseline chromatin profiles using bulk ATAC-Seq. The analysis revealed significantly less open chromatin within promoter regions in the obese group relative to the lean group (*Figure 4A*). Genes regulated by promoters that were less accessible in the obese group were primarily involved in 'myeloid cell activation', 'antigen processing and presentation', and 'neutrophil degranulation' (*Figure 4B*), which included proinflammatory *CD55*, *ITGAL*, and *CCL3L* (*Figure 4C*). In contrast, the few promoters that were more open in the obese group overlapped genes with regulatory roles such as 'regulation of leukocyte activation' and 'chronic inflammatory response' (*Figure 4B*), which included *IL10* and *TFGBI* loci (*Figure 4D*). A direct comparison of baseline chromatin accessibility differences with previously reported transcriptional responses to LPS (*Sureshchandra et al., 2017*) revealed limited overlap. However, maternal obesity associated increase in *IL10* and a decrease in *CD55* promoter accessibility correlated with their transcript levels post LPS stimulation (*Figure 4E*).

We next measured baseline differences in histone modifications that could potentially explain the differences in chromatin accessibility. Nuclear extracts from purified UCB monocytes were probed for specific methylation and acetylation signatures on histone H3K4 and H3K9 using ELISA. No differences in promoter-associated H3K4 mono-, di-, or tri-methylation were observed (*Figure 4F*). While H3K9 acetylation levels were comparable between the groups, heterochromatin-associated H3K9 trimethylation was significantly elevated in UCB monocytes from the obese group (*Figure 4G*). We confirmed these differences by chromatin immunoprecipitation followed by sequencing (ChIP-Seq) of

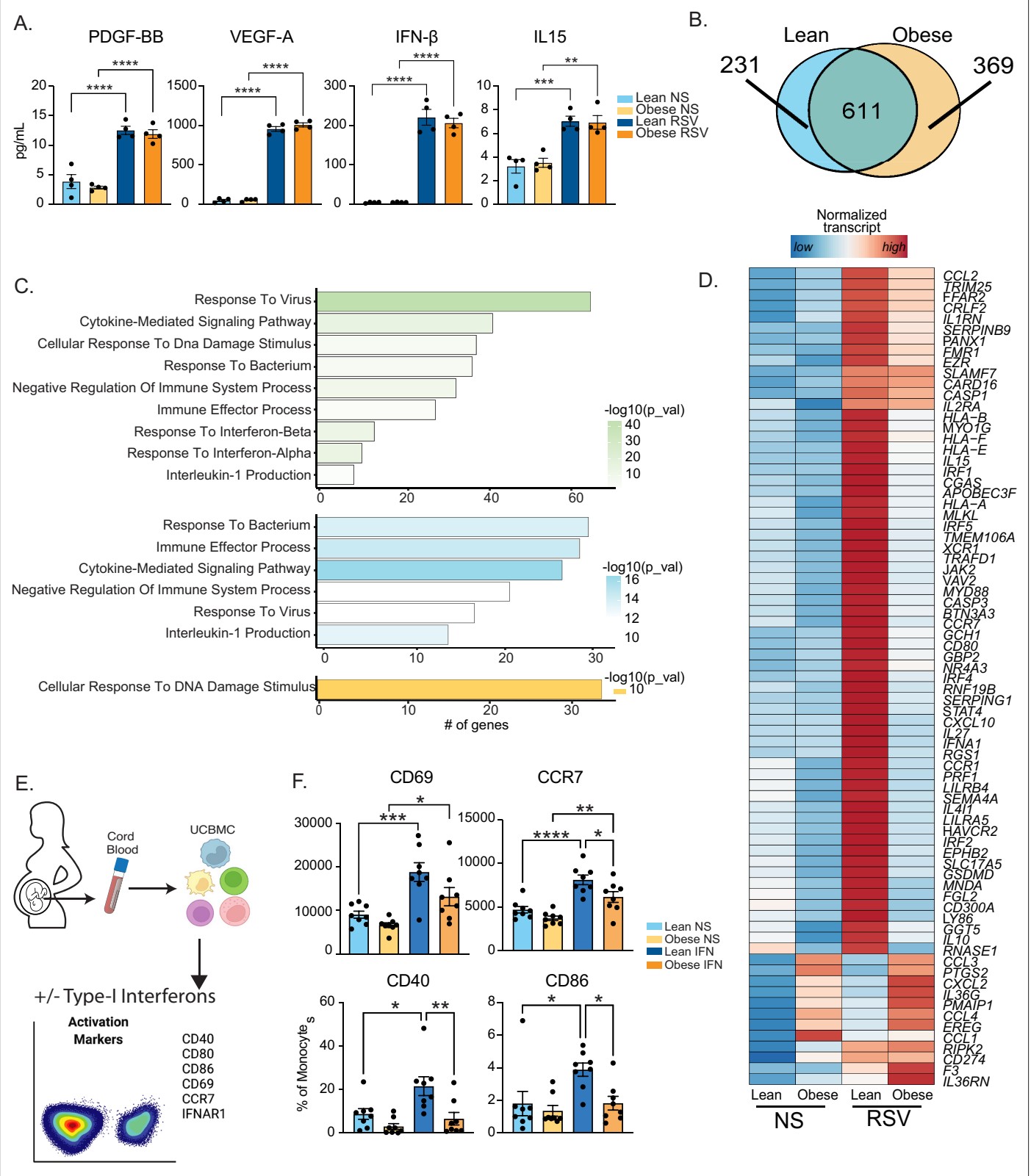

**Figure 3.** Maternal obesity does not alter acute anti-viral responses by cord blood monocytes but dampens responses to type-I interferon. (**A**) Bar graphs comparing levels (mean and ± SEM) of key secreted factors by purified UCB monocytes in response to RSV infection measured using Luminex (n=4/group) compared to no stimulation (NS). (**B**) Venn diagram comparing DEG upregulated in response to RSV infection relative to no stim controls in lean and obese groups. (**C**) Functional enrichment of differentially expressed genes (DEG) detected in both groups (green, top), lean only (blue, middle),

*Figure 3 continued on next page*

*Figure 3 continued*

and obese group only (yellow, bottom) predicted using Metascape. (**D**) Heatmap comparing normalized transcript counts (blue – low to red – high) of genes exclusively upregulated in the lean group following RSV infection, mapping to GO terms 'antigen processing and presentation', 'negative regulation of cell differentiation', and 'myeloid cell differentiation'. (**E**) Experimental design for testing monocyte responses to type-I IFN (n=8/group). UCBMC were stimulated with a mix of human IFNα and IFNβ for 16 hr and activation markers were measured using flow cytometry (n=7–8/group). (**F**) Bar graphs comparing activation markers on cord blood monocytes in response to type-I IFN. *- p<0.05, **- p<0.01, ***- p<0.001, ****-p<0.0001.

The online version of this article includes the following figure supplement(s) for figure 3:

**Figure supplement 1.** Cord blood monocyte responses to RSV.

H3K9 trimethyl regions in purified UCB monocytes, with 5612 peaks (vs. 3888) enriched in the obese group compared to the lean group (*Figure 4H*). Roughly 11% of these repressive marks enriched in the obese group overlapped promoters compared to 1.9% in the lean group (*Figure 4I*). Key genes regulated by impacted promoter mapped to key processes such as 'immune system development' (*PTPN6, VEGFA, TNFSF13B, MAPK3*), 'immune effector process' (*IRF7, IRF8, FCER1G*), and 'NF-kB signaling' (*TRAF1, TFRC, PRKD1*) (*Figure 4J*).

## Epigenetic constraints to UCB monocyte activation with maternal obesity

We next asked if signaling defects downstream of TLR activation contribute to functional differences in UCB monocyte responses with maternal obesity. We began by measuring LPS-induced early phosphorylation events using flow cytometry. No differences in induction of MAPK-p38 or loss of IKBa phosphorylation signals were observed (*Figure 5A*). However, the increase in phosphorylation levels of NF-kB subunit p65 post LPS stimulation was less significant in the obese group (*Figure 5A*). To probe the impact of this modest difference, we measured nuclear NF-kB translocation using imaging flow cytometry. Fewer cells with translocated p50 were detected 4 hr following LPS stimulation (*Figure 5B*).

We next asked if maternal obesity was associated with a dampened epigenetic remodeling in response to immune activation. Loss of the repressive of histone H3K9 trimethylation following LPS stimulation was reduced in UCB monocytes from the obese group (*Figure 5C*). Since metabolic changes precede epigenetic changes, we next measured the overall increase in ECAR (a readout for glycolysis) following 1 hr LPS stimulation using a glycolytic stress assay. Following glucose injection, levels of ECAR increased significantly less in the obese group compared to the lean group (*Figure 5D*). These differences were observed despite the lack of differences in baseline metabolic demands of fetal monocytes as evidenced by both surface expression of primary glucose transporter GLUT1 (*Figure 5—figure supplement 1A*) and overall uptake of extracellular glucose (*Figure 5— figure supplement 1B*) remaining comparable between the two groups.

We next asked if suboptimal changes in chromatin accessibility correlate with attenuated responses to TLR activation in UCB monocytes. First, we compared chromatin accessibility via bulk ATAC-Seq on purified UCB monocytes following 4 hr LPS stimulation (*Figure 5—figure supplement 1C*). As observed in the resting state, fewer open regions were detected in the obese group following LPS stimulation (*Figure 5E*), primarily surrounding the transcription start sites (*Figure 5F*). Motif analysis of these differential peaks revealed enrichment of AP-1 and NF-kB binding sites (*Figure 5G*) including promoters of LPS-induced inflammatory molecules such as *CXCL8, IL6ST, TNFAIP3*, and transcription factor *IRF7* (*Figure 5H* and *Figure 5—figure supplement 1D*). We integrated these differential chromatin signatures with transcriptional signals detected exclusively in the lean groups following LPS, *E. coli*, and RSV stimulation (*Figure 5I*). As expected, LPS-associated changes in chromatin accessibility showed the most overlap with LPS stimulated genes (46 genes) including signaling molecules such as *SOCS1, MAP3K13*, cytokine *IL23A* and chemokine *CXCL8* and to a lesser degree with *E. coli* responsive genes (14 genes). Furthermore, 34 genes that were significantly closed in the obese group were responsive to RSV only in the lean group – including transcription factors *IRF1, IRF2*, MHC class I molecules *HLA-A, HLA-B*, and chemokines *CXCL2, CCL3* (*Figure 5I*).

To capture differences in the heterogeneity of epigenetic response to LPS, we used single-cell ATAC-Seq of resting monocytes or following 4 hr LPS stimulation (*Figure 5—figure supplement 1E*). We identified four major clusters including a cluster of activated monocytes (*Figure 5J*, cluster 1) at baseline, as observed in scRNA-seq (*Figure 1B*). Cells within this cluster had significantly more

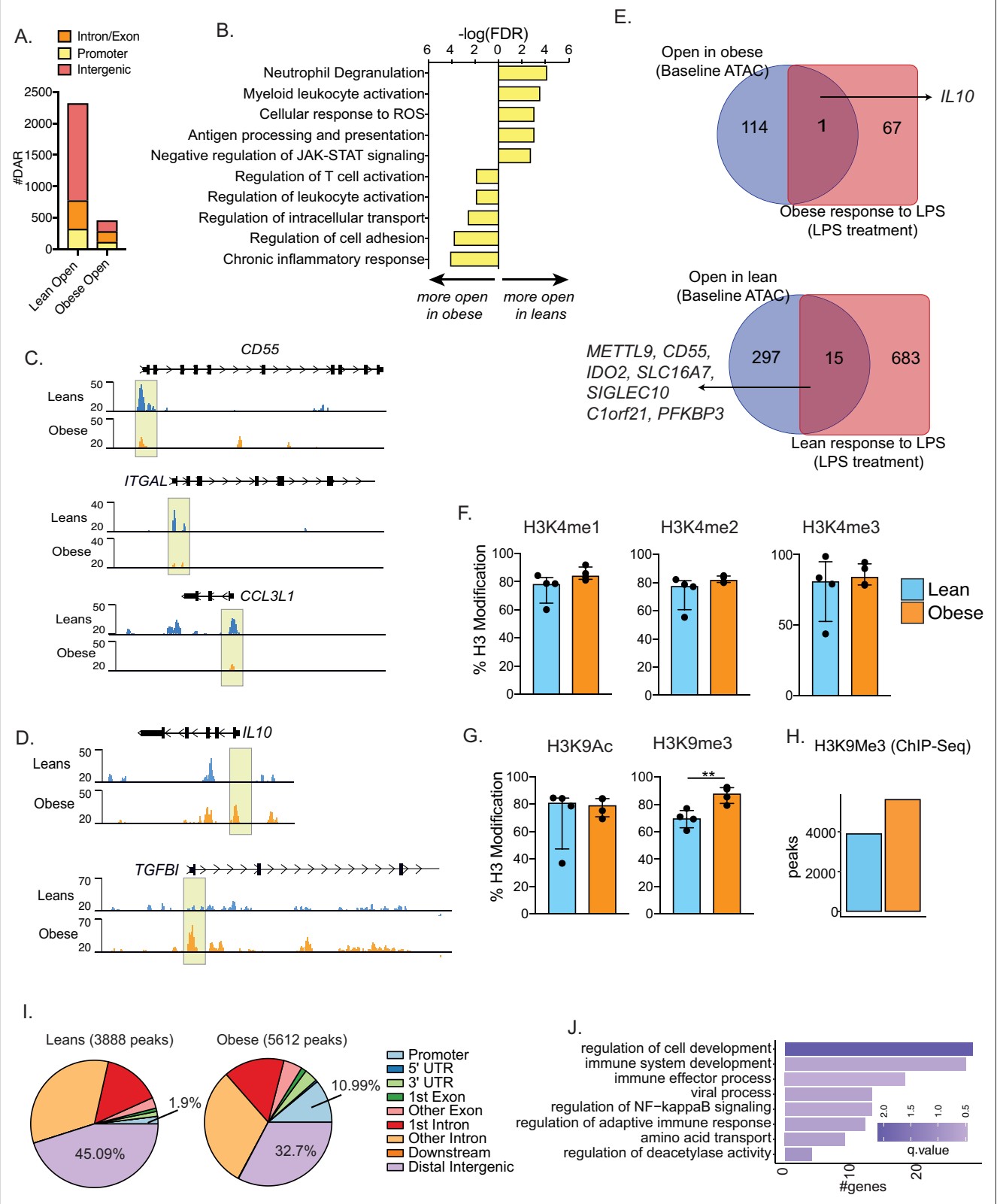

**Figure 4.** Epigenetic priming of cord blood monocytes with maternal obesity. (**A**) Stacked bar graphs comparing genomic contexts of differentially accessible regions (DARs) open in one group relative to the other. (**B**) Functional enrichment of genes regulated by DARs overlapping promoters using Metascape. (**C–D**) Representative pileups of genes more accessible in (**C**) lean or (**D**) obese groups. (**E**) Venn diagrams comparing genes overlapping differentially accessible regions at baseline with gene expression changes following overnight LPS stimulation previously described in this cohort.

*Figure 4 continued on next page*

*Figure 4 continued*

(**F–G**) Bar graphs comparing key histone modifications in (**F**) Histone H3 lysine 4 residue and (**G**) Histone H3 lysine 9 residues from nuclear extracts of resting UCB monocytes (n=3–4/group). Y-axis represents the percentage signal relative to the total H3 signal detected. **- p<0.01.(**H**) Bar graph comparing the number of differential H3K9 trimethyl peaks between lean and obese groups at baseline. (**I**) Context specific annotation of differential H3K9me3 peaks in lean and obese group. (**J**) Functional enrichment of promoters associated H3K9me3 peaks enriched in obese group. Color of the bar is indicative of statistical significance of enrichment.

open promoters that regulate inflammatory genes such as *IL6*, *IL1B*, *CCL2*, and *HLA-DRA* (*Figure 5K*). We observed a twofold increase in the frequencies of this activated monocytes cluster in the lean group but no change in the obese group following LPS stimulation (*Figure 5L*). Moreover, as seen with bulk ATAC-seq, overall changes in chromatin accessibility profiles in response to LPS were more robust in the lean group within all monocyte clusters (*Figure 3—figure supplement 1F*). This included promoter regions of early inflammatory cytokine loci *IL1B*, chemokine *CCL2,* and alarmins *S100A9* and *S100A12* (*Figure 5M*).

## Poor fetal monocyte responses are recapitulated in a non-human primate model of western style diet (WSD)-induced maternal obesity

It is unclear if maternal obesity exerts the same impact on fetal monocytes cells in utero and if this hyporesponsive phenotype extends to tissue-resident macrophages. To address this, we leveraged access to samples from a rhesus macaque model of western-diet-induced obesity (*Figure 6A*). We analyzed fetal PBMCs, splenic, and gut (ileal) macrophages from rhesus macaques obtained at gestational day (GD) 130 from lean dams fed a control chow (CHOW) or obese dams fed a western diet (WSD) (*Figure 6A*). The average length of gestation is 166 days in rhesus macaques, therefore a GD130 time point is considered mid third trimester. Despite a lack of difference in the frequency of circulating monocytes, the frequency of TNFα+IL-6+producing monocytes obtained from the WSD group was significantly reduced following LPS stimulation (*Figure 6B*) as we reported previously (*Sureshchandra et al., 2017*) and *Figure 2—figure supplement 1* for human UCB monocytes.

As described for peripheral monocytes, frequencies of macrophage in fetal spleen or ileum did not vary with maternal diet (*Figure 6C*). Furthermore, surface markers associated with inflammatory (HLA-DR) (*Figure 6D*) or regulatory (CD206, CD209) (*Figure 6E*) were comparable between CHOW and WSD groups. However, when stimulated with *E. coli* for 16 hr, cytokine responses of purified splenic and ileal macrophages were significantly dampened in WSD group (*Figure 6F–G*). Specifically, following stimulation, macrophages from the CHOW group produced significant increases in the levels of canonical Th1 polarizing factors (TNFα, IL-12, type I IFN, GM-CSF), Th17 factors (CCL20), and innate immune cell recruiting factors (CXCL2, IL-8, CXCL11) (*Figure 6F–G*). Moreover, secreted levels of TNFα and CCL20 by fetal ileal macrophages (*Figure 6F*), and levels of both proinflammatory (TNFα, IL-8, IFNα) and regulatory (IL-10) factors by fetal splenic macrophages (*Figure 6G*) were significantly lower with WSD. These data highlight that maternal diet induced obesity alters both mid-gestational circulating fetal monocytes and tissue-resident macrophages responses to pathogens in primates.

## Discussion

Maternal obesity is an independent risk factor for infections during (*Robinson et al., 2005*) and post pregnancy (*Anderson et al., 2013*). Furthermore, neonates born to obese mothers are at high risk for respiratory infections, sepsis, wheezing, and asthma (*Rastogi et al., 2015*; *Suk et al., 2016*). Recent studies have reported dampened responses by human UCB monocytes of babies born to mothers with obesity (*Wilson et al., 2015*; *Sureshchandra et al., 2017*; *Anderson et al., 2021*; *de Goede et al., 2017*) as well as by splenocytes from pups born to obese dams to LPS stimulation (*Sureshchandra, 2021a*). Additional studies indicate increased airway reactivity and aberrant responses to RSV in pups born to dams fed a HFD during gestation (*Griffiths et al., 2016*). These observations align with clinical reports of increased admissions to neonatal intensive care units due to increased incidence of bacterial sepsis and enterocolitis (*Godfrey et al., 2017*; *Blomberg, 2013*; *Myles, 2014*). Mechanisms underlying this dysregulated response are emerging and highlight the role of epigenetic rewiring in line with the Developmental Origin of Health and Disease (DOHaD) theory (*Sureshchandra et al., 2019*; *Peterson et al., 2020*; *Opsahl et al., 2021*; *Nakandakare et al., 2021*; *Arima and*

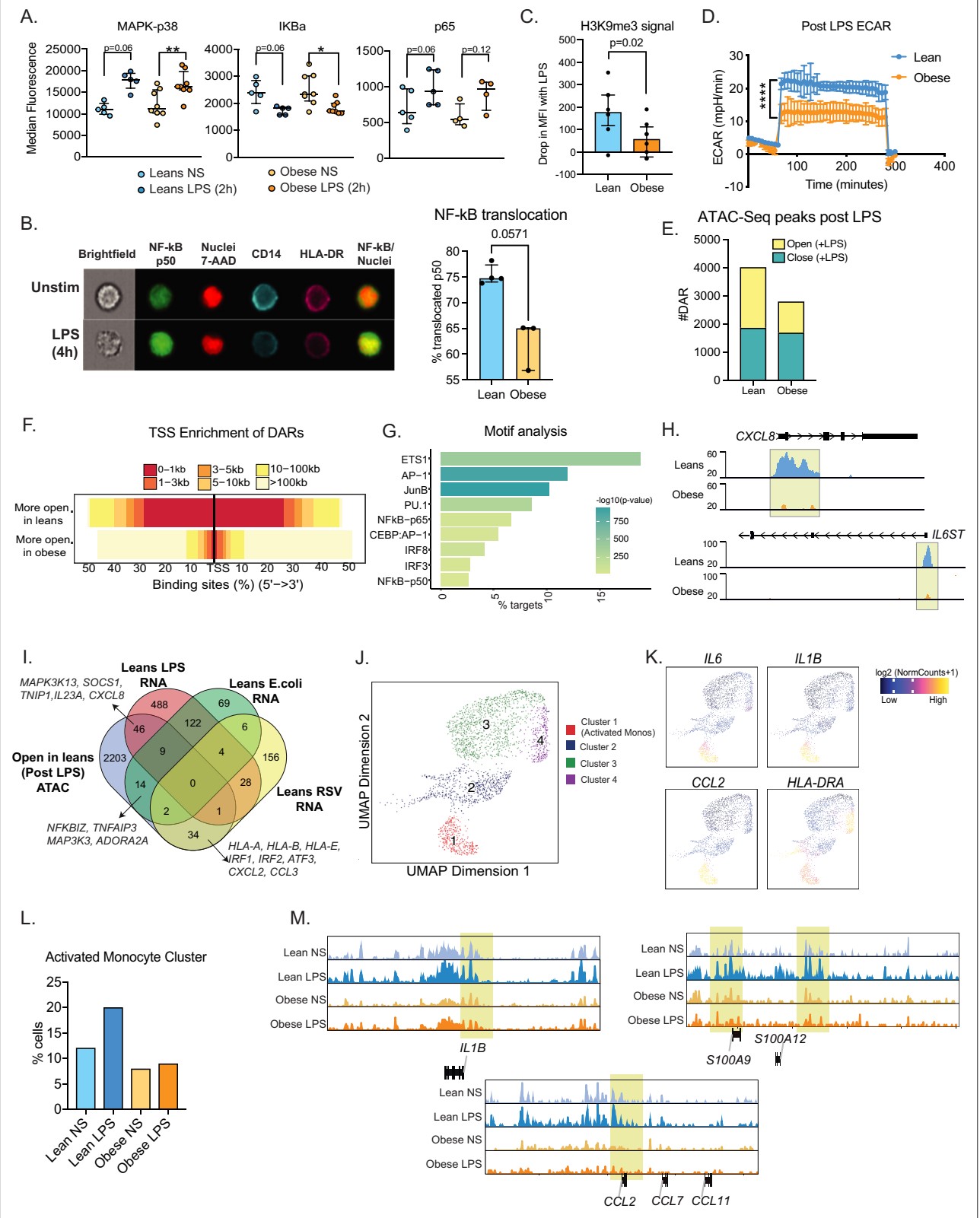

**Figure 5.** Epigenetic constraints to LPS stimulation in cord blood monocytes from babies born to mothers with obesity. (**A**) Dot plots comparing median fluorescence intensity (MFI) ± SEM of phosphorylated signaling molecules downstream of TLR4 sensing (n=5/group). *- p<0.05, **- p<0.01. (**B**) Representative brightfield and fluorescent images of stimulated and unstimulated UCB monocytes profiled using imagine flow cytometry (n=3–4/group). NF-kB (p50–AF488) and nucleus (7-AAD) are shown in green and red respectively. Surface stains for CD14 and HLA-DR are shown in aqua and

*Figure 5 continued on next page*

*Figure 5 continued*

fuchsia respectively. Overlay of NF-kB and nuclei stain was used to determine translocation within CD14 + HLA-DR+monocytes. Bar graph comparing percentage translocated cells following LPS stimulation in lean and obese groups. (**C**) Bar graph comparing changes in trimethyl modification on H3K9 residues following LPS stimulation (relative to no stimulation controls) detected using flow cytometry. (**D**) Graph representing the kinetics of ECAR of stimulated monocytes following glucose injection and blockade of glycolysis. (**E**) Bar graph comparing DAR frequencies in each group following LPS stimulation. (**F**) Heatmap demonstrating overall accessibility differences following LPS stimulation around the promoter. (**G**) Over-represented transcription factors identified from motif analysis of DARs more open in lean compared to obese groups following LPS stimulation. X-axis represented percentage of peaks with motifs identified and color represents p-value on log10 scale (**H**) Pileups of key inflammatory loci post LPS stimulation (**I**) Four-way Venn of genes accessible (ATAC-Seq) exclusively in the lean group post LPS stimulation with genes exclusively upregulated (RNA-Seq) in the lean group following LPS, *E. coli* and RSV infection. Select overlapping genes are highlighted. (**J**) UMAP of single nuclei ATAC-Seq of LPS stimulated and sorted monocytes. (**K**) Feature plots demonstrating a cluster of activated monocytes (cluster 1). Color intensity represents fragments mapping to open chromatin regions. (**L**) Proportions of monocytes within each group mapping to activated monocyte cluster. (**M**) Pileups of inflammatory loci in activated monocytes with/without LPS stimulation. * - p<0.05; ** - p<0.01.

The online version of this article includes the following figure supplement(s) for figure 5:

**Figure supplement 1.** Epigenetic responses to LPS in cord blood monocytes.

---

*Fukuoka, 2020*; *Sureshchandra et al., 2021b*; *Sureshchandra et al., 2018*; *Wei et al., 2021*). In this study, we confirm and extend these earlier observations using an independent cohort of cord blood samples. We used a combination of functional and genomic assays to demonstrate that defective responses to LPS extend to bacterial and viral pathogens and are regulated both at metabolic and epigenetic levels.

We leveraged scRNA-seq to capture the transcriptional heterogeneity of cord blood monocytes and the impact of maternal obesity on UCB monocyte cell states. We identified a subset of UCB monocytes that express high levels of transcript associated with inflammatory processes suggesting that these cells are poised to respond to stimulation. This subset was nearly absent in the obese group. Furthermore, several pathogen-sensing pathways were downregulated in resting UCB monocytes with maternal obesity, suggesting a general state of immune tolerance. This conclusion is congruent with the dampened transcriptional response to LPS reported previously (*Wilson et al., 2015*; *Sureshchandra et al., 2019*) and the reduced migratory capacity of UCB monocytes from babies born to mothers with obesity. Importantly, these data indicate that UCB monocytes from babies born to mothers with obesity are predisposed toward a refractory/immune-tolerant state.

To extend the previous studies carried out with LPS, here, we measured ex vivo responses to bacterial (*E. coli*) and viral (RSV) pathogens since high maternal BMI was linked to increased risks of severe bacterial and respiratory tract viral infection in the offspring (*Myles et al., 2013*; *Rajappan et al., 2017*). Following *E. coli* infection, we report dampening of transcriptional and cytokine responses while expression of negative regulators was increased in UCB monocytes with maternal pregravid obesity. A similar phenotype was recorded in a previous study where UCB monocyte-derived macrophages from mothers with obesity displayed an unbalanced response to M1 and M2 polarizing stimuli mediated by alteration in methylation patterns at key promoter regions (*Cifuentes-Zúñiga et al., 2017*). Surprisingly, in our study, we observed that ex vivo phagocytic ability was enhanced with obesity. Higher phagocytic affinity is a salient feature of regulatory monocytes/macrophages (*Schulz et al., 2019*; *Jaggi et al., 2020*); hence this observation supports our proposed argument of regulatory skewing with maternal obesity. Collectively, these findings provide a potential explanation for increased susceptibility to bacterial infections with maternal obesity in both mouse models (*Myles et al., 2013*) and neonates in the clinic (*Castaneda et al., 2022*; *Yang et al., 2019*). In line with earlier data from animal studies (*Griffiths et al., 2016*), we report a poor induction of key genes such as interferon response factors (IRF), co-stimulation molecules, chemokine receptors, and numerous ISG following ex vivo RSV infection.

Intriguingly, poor induction of ISG occurred despite comparable secretion of IFNβ by RSV stimulated UCB monocytes. It is possible that lower transcript levels of *IFNA* could contribute to this outcome. However, other factors such as epigenetic or signaling defects are likely to contribute given the dampened ability of the cells to respond to IFNα/IFNβ when supplied in vitro despite the comparable surface protein expression of IFNAR1. Similarly, surface protein expression of TLR key signal transduction proteins downstream of TLR4 was comparable between the two groups despite lower inflammatory responses by the obese group (*Sureshchandra et al., 2017*). Here we show that nuclear

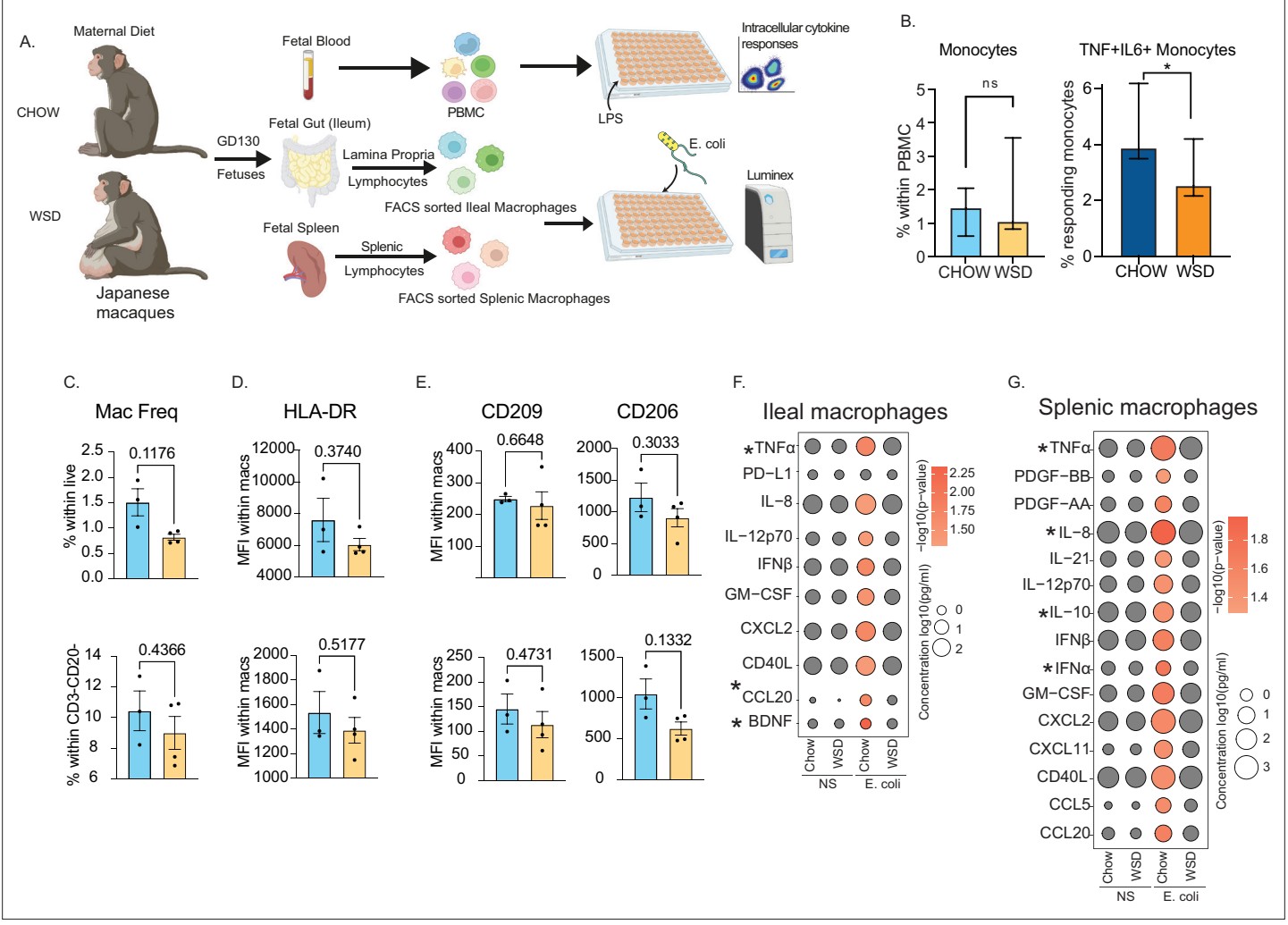

**Figure 6.** Maternal WSD attenuates fetal monocyte and macrophage cytokine responses to ex vivo stimulation. (**A**) NHP model of maternal diet-induced obesity. Fetal (GD130) PBMC, ileal leukocytes, and spleenocytes were obtained from rhesus macaque born to dams on a control chow diet (n=3) and western-style diet (n=4). Fetal macrophages from ileal lamina propria lymphocytes and splenocytes were FACS-sorted and cultured overnight with *E. coli*. Supernatants were collected and cytokines/chemokines were measured using Luminex. (**B**) Bar graph representing the frequency (mean and ± SEM) of responding fetal monocytes to LPS in PBMC isolated from GD130 fetal circulation measured by intracellular cytokine staining. (**C**) Frequencies of CD14 + HLA-DR+macrophages within live total ileal leukocytes (top) or live CD3-CD20- leukocytes (bottom) (**D–E**) Median fluorescence intensity (MFI) of (**D**) M1-associated HLA-DR and (**E**) M2-associated CD209 and CD206 within live macrophage population in ileum (top) and spleen (bottom). (**F–G**) Bubble plots comparing significantly different cytokine/chemokine levels in response to *E. coli* stimulation in (**F**) gut (ileum) and (**G**) splenic macrophages. Statistical differences between chow and WSD groups are highlighted. *-p<0.05.

translocation of the key transcription factor NF-kB was reduced with maternal obesity. While chromatin accessibility regulates NF-kB binding and subsequent induction of gene expression, NF-kB binding also influences the chromatin state by recruiting chromatin-modifying enzymes and evicting negative regulators (*Bhatt and Ghosh, 2014*). Indeed, our data suggest an increased abundance of a repressive histone mark (H3K9me3) that persisted upon LPS stimulation in the obese group. Furthermore, this repressive mark preferentially overlapped promoters of genes involved in immune signaling, suggesting compaction of coding regions in the obese group relative to the leans. Unbiased genome accessibility analyses supported this hypothesis, with regions closed in the obese group, primarily overlapping promoters of key cytokine and inflammatory genes.

Interestingly, in addition to reduced translocation of NF-kB in the obese group, we observed poor induction of epigenetic responses to LPS. Compared to the lean group, cells from the obese group had limited reduction in global levels of the repressive H3K9 trimethyl mark and modest changes in chromatin accessibility following LPS stimulation. Interestingly, the regions more closed in the

obese group harbored binding sites for AP-1 and NF-kB, which cooperatively work downstream of TLR4 and regulate the expression of several effector molecules. Single-cell RNA and ATAC-Seq highlighted that these defective responses might be restricted to the small subset of activated UCB monocytes, expressing high levels of proinflammatory molecules such as *IL6*, *IL1B* identified by our scRNA-Seq analysis and confirmed by scATAC-Seq, which fail to expand with LPS stimulation in the obese group. Collectively, these findings suggest that UCB monocytes from obese group are unable to effectively integrate signals downstream of PRR, which culminate into dysregulated opening of chromatin, nuclear translocation of transcription factors, and their binding to specific genomic loci. Closed chromatin profiles at baseline suggest that these cells are epigenetically poised to respond poorly to subsequent stimulation. Thus, the lack of robust metabolic, epigenetic, and transcriptional response to stimulation provides further evidence for mechanistic constraints contributing to innate immune tolerance.

The transgenerational impact of maternal obesity on the epigenome of offspring's immune system has been long appreciated but is hard to model in humans. Several studies have highlighted alterations in methylome and chromatin accessibility landscape of the offspring's circulating immune cells that are evident as early as at birth (*Wilson et al., 2015*; *Sureshchandra et al., 2021c*; *Sureshchandra et al., 2019*; *Sureshchandra et al., 2021b*; *Keleher et al., 2018*). We previously reported that maternal obesity is associated with differentially methylated regions that regulate metabolism, cell migration and adhesion of myeloid cells (*Sureshchandra et al., 2017*). Some of these epigenetic modifications may be inherited through alterations in gametes, but others may be in response to maternal inflammatory and nutritional cues (*Li, 2018*). Surprisingly, among several circulating factors tested, we only observed elevated levels of leptin in UCB plasma of mothers with obesity. Altered leptin signaling in utero may predispose the fetus to leptin resistance and possibly be the link to the high rates of fetal overgrowth, childhood obesity, and the risk of developing asthma later in childhood (*Yura et al., 2005*; *Stefaniak et al., 2019*; *Castro-Rodriguez et al., 2020*). Furthermore, leptin has been shown to stimulate monocyte activation and differentiation (*Santos-Alvarez et al., 1999*).

A major limitation of studies in humans is our inability to interrogate the burden of maternal obesity on peripheral blood monocytes and tissue resident macrophages. In this study, we addressed this limitation by analyzing circulating monocytes and tissue-resident macrophages obtained from third-trimester rhesus macaque fetuses fed either control or WSD. As observed in human umbilical cord blood samples, we report dampened responses to LPS and *E. coli*. This observation provides evidence that maternal diet playing a central role in utero re-programming of the developing immune system. More importantly, the fact that both circulating monocytes and tissue-resident macrophages generated a dampened response strongly suggests that this programming occurs early during gestation. Indeed, monocytes and macrophages are some of the earliest immune cells to develop with the first wave originating from late yolk-sac-derived erythromyeloid progenitors and fetal liver (*Wolf et al., 2019*). A recent study demonstrated a proinflammatory phenotype within bone marrow macrophages of GD130 fetuses born to dams on WSD (*Sureshchandra et al., 2022*). The heightened maternal inflammatory milieu associated with obesity/WSD during critical windows of fetal immune ontogeny may skew the maturation of immune cells toward a regulatory state, perhaps a protective adaptation to limit fetal inflammation. However, this regulatory skewing also contributes to less-than-ideal responses against pathogens, as observed in both animal models and human studies. Future studies will address outstanding critical questions regarding the persistence of this phenotype during childhood and the impact of reversal of maternal diet on the restoration of immune function in the offspring.

## Acknowledgements

We thank Mr. Allen Jankeel and Ms. Gouri Ajith for help with immune assays and the preparation of RNA-Seq libraries. We thank Dr. Jennifer Atwood from the flow cytometry core at the Institute for Immunology, UCI for assistance with sorting experiments and imaging flow cytometry. We thank Dr. Melanie Oakes from UCI Genomics and High-Throughput Facility for assistance with 10 X library preparation and sequencing.

This study was supported by grants from the National Institutes of Health 1K23HD06952 (NEM), R03AI112808 (IM), 1R01AI142841 (IM), and 1R01AI145910 (IM).

## Additional information

### Funding

| Funder | Grant reference number | Author |
|---|---|---|
| National Institute of Allergy and Infectious Diseases | R03AI112808 | Ilhem Messaoudi |
| National Institute of Allergy and Infectious Diseases | 1R01AI142841 | Ilhem Messaoudi |
| National Institute of Allergy and Infectious Diseases | 1R01AI145910 | Ilhem Messaoudi |
| National Institute of Allergy and Infectious Diseases | 1K23HD06952 | Nicole E Marshall |

The funders had no role in study design, data collection and interpretation, or the decision to submit the work for publication.

### Author contributions

Suhas Sureshchandra, Conceptualization, Data curation, Formal analysis, Visualization, Writing - original draft, Writing - review and editing; Brianna M Doratt, Data curation, Formal analysis, Visualization, Writing - review and editing; Norma Mendza, Data curation, Formal analysis; Oleg Varlamov, Resources; Monica Rincon, Resources, Data curation; Nicole E Marshall, Conceptualization, Supervision, Funding acquisition, Writing - original draft, Writing - review and editing; Ilhem Messaoudi, Conceptualization, Resources, Data curation, Formal analysis, Supervision, Funding acquisition, Methodology, Writing - original draft, Project administration, Writing - review and editing

### Author ORCIDs

Brianna M Doratt ⓘ http://orcid.org/0000-0002-8107-724X
Monica Rincon ⓘ http://orcid.org/0000-0001-5574-585X
Ilhem Messaoudi ⓘ http://orcid.org/0000-0003-3203-2405

### Ethics

This study was approved by the Institutional Ethics Review Board of Oregon Health and Science University (STUDY00020735 "Perinatant Early Determinants of Immune Development") and the University of California, Irvine (protocol number 2017-3397 "Impact of maternal pre-pregnancy obesity on the offspring immune system"). Written consent was obtained from all subjects.

### Decision letter and Author response

Decision letter https://doi.org/10.7554/eLife.81320.sa1
Author response https://doi.org/10.7554/eLife.81320.sa2

## Additional files

### Supplementary files
- MDAR checklist
- Supplementary file 1. Donor contributions to scRNAseq UMAP .

### Data availability

The datasets supporting the conclusions of this article are available on NCBI's Sequence Read Archive PRJNA847067 and PRJNA914662.

The following datasets were generated:

| Author(s) | Year | Dataset title | Dataset URL | Database and Identifier |
|-----------|------|---------------|-------------|-------------------------|
| Messaoudi I | 2023 | Maternal obesity blunts antimicrobial responses in fetal monocytes | http://www.ncbi.nlm.nih.gov/bioproject/?term=PRJNA847067 | NCBI BioProject, PRJNA847067 |
| Messaoudi I | 2023 | Maternal obesity blunts antimicrobial responses in fetal monocytes | http://www.ncbi.nlm.nih.gov/bioproject/?term=PRJNA914662 | NCBI BioProject, PRJNA914662 |

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
