## [Editor Report]

This manuscript will be of interest to a broad range of researchers studying immunology, obesity and metabolism, links between maternal health and pathophysiological responses in the offspring. The comprehensive studies using RNA-seq, scRNA-seq, ATAC-seq, scATAC-seq and CUT& Tag represent an important resource for understanding the transcriptomic and epigenetic shifts in the monocytes of newborns. The experiments involving stimulation of monocytes with pathogens offer convincing evidence for the dysfunction of monocytes in the newborn. The analysis of fetal macrophages of non-human primates are of fundamental importance for the field.

---

## [Decision Letter]

**Decision letter after peer review:**

Thank you for submitting your article "Maternal obesity blunts antimicrobial responses in fetal monocytes" for consideration by *eLife*. Your article has been reviewed by 3 peer reviewers, including Jalees Rehman as the Reviewing Editor and Reviewer #1, and the evaluation has been overseen by a Reviewing Editor and Arturo Casadevall as the Senior Editor. The following individual involved in review of your submission has agreed to reveal their identity: Siroon Bekkering (Reviewer #3).

The reviewers have discussed their reviews with one another, and the Reviewing Editor has drafted this to help you prepare a revised submission. The reviewers and editors agree that your research question is of significant interest to the field

Essential revisions:

1) Clarify and discuss how the mode of delivery could impact epigenetic + transcriptomic shifts; a sensitivity analysis of the mode of delivery and potential differences as its consequence would be important

2) Was the viability of the cord blood samples assessed after thawing cells? Differences in sample processing and thawing can profoundly affect transcriptomic data.

3) The analysis of monocyte subsets does not follow monocyte classifications that have been used by other groups in the field. Re-analysis of monocyte subsets according to recommendations made by reviewers in order to contextualize the monocyte subset findings with the existing literature

4) The chromatin accessibility data (ATAC) need to be integrated with the mRNA data. An in-depth bioinformatic analysis is required to assess whether there is concordance or difference between the genes that are differentially accessible via ATAC-seq and those that are differentially expressed via RNA-seq. Studies on differences in histone methylation need to integrated with differences in chromatin accessibility and mRNA levels. Global changes in histone methylation are difficult to interpret. ChIP-qPCR at defined sites would be more relevant.

5) There is a need for re-analysis of the scRNA-seq data with the addition of information on population frequencies, patient-specific differences/similarities, clear identification of cell types that are affected. The reviewer comments provide guidelines for how to strengthen the bioinformatic analysis.

6) The increase in *E. coli* uptake of obesity-associated monocytes is counterintuitive and sufficient interpretation has to be provided.

7) The macaque model is very powerful but is presented in such a sparse manner that its relevance and impact are not fully clear. It is among the most innovative features of this manuscript but the data need to be presented more comprehensively to justify the conclusions. The PCA analysis is not interpretable. Instead, it is important to compare the gene expression changes between the macaque model and the cord blood gene expression data from humans.

8) The macaque study and the ethical rationale + approval for conducting the macaque study need to be described in more detail.

9) The Discussion section has to highlight the significance of the paper better by outlining how it is different from the published literature. Epigenetic and transcriptomic changes in offsprings of obese mothers have been previously described, but an in-depth analysis of the scRNA-seq and ATAC-seq data would also form the basis of a stronger Discussion. Integrating the expanded analysis of the non human primate model into the Discussion will further strengthen the conclusions and help clarify how this work will advance the field.

*Reviewer #1 (Recommendations for the authors):*

The following revisions could strengthen the paper:

1. Clearly delineate where ATAC-seq and RNA-seq data are concordant versus non-concordant. What is the overlap between the accessible genome regions which can be assigned to specific genes (promoters, coding regions) and the RNA-seq during baseline and with stimulation?

2. The same should also be shown for the scRNA-seq and the scATAC-seq data. Where are the overlaps, where are the differences? In its current presentation, readers are shown a large number of graphs, often selected genes, but it is difficult for the readers to form cohesive assessments of the transcriptomic and epigenetic states.

3. It would be helpful to explain the increased *E. coli* uptake by obesity-associated monocytes. Based on the idea of the impaired immune responses in the offspring of obese mothers, one would expect the opposite. Is this perhaps a function of uptake without bacterial killing? Are the lean offspring better at elimination? A time assay could help address this. Another key point is that much of the bacterial uptake would be by monocyte derived macrophages and not necessarily the monocytes themselves. Repeating these studies with monocyte-derived macrophages, and performing a time course to distinguish uptake and bacterial killing may resolve this.

4. The PCA in the NHP study (Figure 6C and D) is not very helpful. Instead, it would be much more important to show additional phenotype differences between the fetal macrophage groups, such as flow cytometry markers and bacterial uptake.

*Reviewer #2 (Recommendations for the authors):*

The manuscript by Sureshchandra et al. investigates the aberrant molecular characteristics of cord blood monocytes from offsprings of obese mothers, following bacterial and viral challenges. By combining transcriptomic, epigenetic and cytokine profiling the authors identify gene expression and epigenetic changes in cord blood monocytes that correlate with putative defects in immune responses.

Orthogonal approaches to identify effects of obesity on mononuclear cells biology. These include single cell transcriptomics to identify and classify cell subpopulations between lean and obese conditions, epigenomic assays including ATAC-seq to identify epigenetic changes and their associated loci, and cytokine measurements as a readout of monocyte functional output.

Macaque model to measure relative early gestation changes that could under further investigation provide valuable insights on how western-diet induced obesity in the mother can mediate changes in the immune responses of early embryos.

Main Critiques

While the authors use several assays to identify monocyte changes that are associated to maternal obesity, extensive further data analysis is required to improve this manuscript.

1. The scRNA datasets are superficially analyzed and key questions regarding the exact cell types that are affected as a result of maternal obesity remain unaddressed. Specifically:

a. Figure 1B indicates the cell types identified by scRNA-seq and 4 patients per group are indicated. A table describing the frequencies of each cell type identified in B is completely missing. The scRNA profiles of the independent patients should be shown in a supplementary figure. Are the results obtained by scRNA-seq in line with the FACS analysis? Figure S1A indicates no significant differences are observed by FACS in monocytes but figure 1D indicates statistically significant changes but it is not clear what these represent.

b. In figure1D it is not clear what the boxplots represent. If these are frequencies of the subclusters for the 4 patients then the four independent patients numbers should be shown. In the classical monocytes graph p=0.1 is probably wrong.

c. It is unclear how the genes in 1E and F were selected. What are the DEG identified from the single cell RNA seq data when obese and lean patients are compared for each identified subcluster?

d. The module scores in 1G should be plotted on the actual single cell map to verify that these signatures are only obtained/different in Monocyte populations and are not driven by any other cell type profiled.

2. A comparison between the viral and *E. coli* responses is missing. A commentary for the similarity/disparity between these responses could be of potential interest.

3. Epigenetic data are presented in non-standard manner that makes interpretation of data quality and conclusion difficult to interpret.

a. Average profiles of ATAC-seq signals around the TSS for the two groups should be performed following depth normalization to ensure that the profiles look similar and are comparable between groups. The signal at differential sites should also be shown and replicates should be indicated to ensure robustness of the results. What are the identified motifs of transcription factors associated with the differential regions – Is NfKb identified given that a significant role is attributed in figure 5.

b. Histone Elisa and Histone Flow are not the golden standard to indicate epigenetic/chromatin changes. ChIP (PCR or Seq) is required to be performed to identify (1) locations in which changes occur and (2) identify the affected histone mark. It is unclear what is the interpretation of the differences observed in H3K9me3 -a repressive mark that is predominantly distributed in pericentromeric heterochromatin.

4. The PCA plots in Figure6 are uninterpretable. What is the meaning of the circles in figure 6C and D and how are were they obtained? They do not appear to support PCA grouping.

*Reviewer #3 (Recommendations for the authors):*

– Since the authors are working with frozen cells, analysis of viability after thawing is essential. There is no mention of viability staining or measurement after thawing anywhere.

Please provide, otherwise this is a major confounder.

– Mode of delivery is a major confounder which can affect the CBMC response. Did the authors check whether this affected their results?

– Could the authors provide the statistics on table 1? Where are the two groups different?

– Please provide a gating strategy.

– The monocyte subset analysis by flow cytometry is not according to recommendations (3 subsets: classical CD14++CD16dim, intermediate CD14++CD16+, non-classical CD14dimCD16++). Why was this chosen and could the authors go back and provide the recommended monocyte classifications?

– Why was ELISA used to determine histone marks? This is very non-specific and only tells you something about the global H3K4 methylation, not gene specific. ChIP-PCR is a relatively easy method and doesn't require expensive sequencing if you already have ATAC-seq data

– Was testing for equal variance performed?

– Outliers were taken out before normality testing. However both ROUT and Grubbs assume a gaussian distribution. Please consider doing the normality tests first and only take out outliers if data are normally distributed.

– Whereas almost the whole manuscript is very well written, the discussion lacks in many things. First of all, the description of the findings is too long, but the comparison to other literature is very short. There is no mention of limitations at all, I cannot imagine the authors cannot think of limitations. And what about future perspectives? How can these findings influence the clinics or future pregnancies? I think the findings are important, so the authors are hereby invited to tell the readers too.

---

## [Author Response]

Essential revisions:1) Clarify and discuss how the mode of delivery could impact epigenetic + transcriptomic shifts; a sensitivity analysis of the mode of delivery and potential differences as its consequence would be important

All experiments were conducted using a mix of samples obtained from C-section and vaginal delivery avoiding a bias towards either. We now include an example in Supp Figure 1 where we show no differences in the UCB monocyte response to LPS based on mode of delivery.

2) Was the viability of the cord blood samples assessed after thawing cells? Differences in sample processing and thawing can profoundly affect transcriptomic data.

Cell viability was assessed after thawing cells. It was comparable between the two groups and always >60% (Supp Figure 2H).

3) The analysis of monocyte subsets does not follow monocyte classifications that have been used by other groups in the field. Re-analysis of monocyte subsets according to recommendations made by reviewers in order to contextualize the monocyte subset findings with the existing literature

We have re-gated all the monocyte data using the gating strategy suggested by reviewer 1 and report those findings in Supp Figure 2.

4) The chromatin accessibility data (ATAC) need to be integrated with the mRNA data. An in-depth bioinformatic analysis is required to assess whether there is concordance or difference between the genes that are differentially accessible via ATAC-seq and those that are differentially expressed via RNA-seq.

We provide a better discussion of the data sets obtained by RNA-Seq and ATAC-Seq in results and Discussion sections.

Studies on differences in histone methylation need to integrated with differences in chromatin accessibility and mRNA levels. Global changes in histone methylation are difficult to interpret. ChIP-qPCR at defined sites would be more relevant.

While limited number of cells precluded us from carrying out ChIP-qPCR, we leveraged a new technology of CUT&Tag to look at genomic contexts associated with H3K9me3. Our new data agrees with our ATAC-Seq data and bolsters our conclusion of a state of transcriptional repression in cord blood monocytes induced by maternal obesity.

5) There is a need for re-analysis of the scRNA-seq data with the addition of information on population frequencies, patient-specific differences/similarities, clear identification of cell types that are affected. The reviewer comments provide guidelines for how to strengthen the bioinformatic analysis.

We now provide additional information on other immune population frequencies and patient-level statistical analysis.

6) The increase in *E. coli* uptake of obesity-associated monocytes is counterintuitive and sufficient interpretation has to be provided.

Higher phagocytic ability is in line with the regulatory phenotype that we observe with maternal obesity. We now provide references that support this statement.

7) The macaque model is very powerful but is presented in such a sparse manner that its relevance and impact are not fully clear. It is among the most innovative features of this manuscript but the data need to be presented more comprehensively to justify the conclusions. The PCA analysis is not interpretable. Instead, it is important to compare the gene expression changes between the macaque model and the cord blood gene expression data from humans.

We didn’t carry out RNA-Seq on the NHP samples and no longer have cells to repeat the stimulations for transcriptional experiments. We have removed the PCA, added the phenotype of the tissue residence macrophages, and expand on the utility of the model.

8) The macaque study and the ethical rationale + approval for conducting the macaque study need to be described in more detail.

All details regarding the monkey studies are provided in reference #60

9) The Discussion section has to highlight the significance of the paper better by outlining how it is different from the published literature. Epigenetic and transcriptomic changes in offsprings of obese mothers have been previously described, but an in-depth analysis of the scRNA-seq and ATAC-seq data would also form the basis of a stronger Discussion. Integrating the expanded analysis of the non human primate model into the Discussion will further strengthen the conclusions and help clarify how this work will advance the field.

The discussion has been completely overhauled to address the concerns raised by the reviewers and editors

Reviewer #1 (Recommendations for the authors):The following revisions could strengthen the paper:1. Clearly delineate where ATAC-seq and RNA-seq data are concordant versus non-concordant. What is the overlap between the accessible genome regions which can be assigned to specific genes (promoters, coding regions) and the RNA-seq during baseline and with stimulation?2. The same should also be shown for the scRNA-seq and the scATAC-seq data. Where are the overlaps, where are the differences? In its current presentation, readers are shown a large number of graphs, often selected genes, but it is difficult for the readers to form cohesive assessments of the transcriptomic and epigenetic states.

We have included an new integration section – with overlap of baseline ATAC-Seq (data from this study) with gene expression responses (from a previous study) following LPS stimulation in lean and obese groups – Figure 4E. Additionally, we report overlap of LPS induced chromatin changes with gene expression changes following LPS, *E. coli* and RSV stimulation in Figure 5I. Collectively, these changes provide the reader with a better link between chromatin accessibility and gene expression differences and their discordance with maternal obesity. With single cell assays, unfortunately, integration with unimodal samples (ATAC and RNA from different donors/cells) is technically challenging. However, we’ve addressed these concerns in both the results and discussion.

3. It would be helpful to explain the increased *E. coli* uptake by obesity-associated monocytes. Based on the idea of the impaired immune responses in the offspring of obese mothers, one would expect the opposite. Is this perhaps a function of uptake without bacterial killing? Are the lean offspring better at elimination? A time assay could help address this. Another key point is that much of the bacterial uptake would be by monocyte derived macrophages and not necessarily the monocytes themselves. Repeating these studies with monocyte-derived macrophages, and performing a time course to distinguish uptake and bacterial killing may resolve this.

*E. coli* uptake assay is a standard way of measuring cellular phagocytosis by flow cytometry. We would like to clarify that despite impaired ex vivo cytokine responses and poor migration, UCB monocytes demonstrate higher ability to phagocytize pathogens. This is counterintuitive but not surprising, given that enhanced phagocytosis is a hallmark of regulatory monocytes/macrophages.

4. The PCA in the NHP study (Figure 6C and D) is not very helpful. Instead, it would be much more important to show additional phenotype differences between the fetal macrophage groups, such as flow cytometry markers and bacterial uptake.

As requested by all reviewers, we have removed PCA from Figure 6. We have now included phenotyping data for ileal and splenic macrophages in Figure 6C-6E, which were collected during cell sorting. Unfortunately, we do not have any additional cells available for killing/bacterial uptake assay.

Reviewer #2 (Recommendations for the authors):The manuscript by Sureshchandra et al. investigates the aberrant molecular characteristics of cord blood monocytes from offsprings of obese mothers, following bacterial and viral challenges. By combining transcriptomic, epigenetic and cytokine profiling the authors identify gene expression and epigenetic changes in cord blood monocytes that correlate with putative defects in immune responses.Orthogonal approaches to identify effects of obesity on mononuclear cells biology. These include single cell transcriptomics to identify and classify cell subpopulations between lean and obese conditions, epigenomic assays including ATAC-seq to identify epigenetic changes and their associated loci, and cytokine measurements as a readout of monocyte functional output.Macaque model to measure relative early gestation changes that could under further investigation provide valuable insights on how western-diet induced obesity in the mother can mediate changes in the immune responses of early embryos.Main CritiquesWhile the authors use several assays to identify monocyte changes that are associated to maternal obesity, extensive further data analysis is required to improve this manuscript.1. The scRNA datasets are superficially analyzed and key questions regarding the exact cell types that are affected as a result of maternal obesity remain unaddressed. Specifically:a. Figure 1B indicates the cell types identified by scRNA-seq and 4 patients per group are indicated. A table describing the frequencies of each cell type identified in B is completely missing. The scRNA profiles of the independent patients should be shown in a supplementary figure. Are the results obtained by scRNA-seq in line with the FACS analysis? Figure S1A indicates no significant differences are observed by FACS in monocytes but figure 1D indicates statistically significant changes but it is not clear what these represent.

While we profiled all live cells within Umbilical cord blood mononuclear cells, the goal of the study was to characterize changes in the myeloid cell compartment. Deep characterization of the CD4 T cell compartment with maternal obesity has been previously described by our lab. Per the reviewer’s request, we have now provided Supp Table 1 and Supp Figure 3D with cluster proportions from each donor. scRNA profiles of each donor is now available in Supp Figure 3C. Supp Figure 1A are total monocyte numbers determined by Complete Blood Count (CBC). Differences observed in Figure 1D is now clear in Supp Figure 3C and 3D.

b. In figure1D it is not clear what the boxplots represent. If these are frequencies of the subclusters for the 4 patients then the four independent patients numbers should be shown. In the classical monocytes graph p=0.1 is probably wrong.

Figure 1D is boxplot of proportions of monocyte subsets within the UMAP of all cells profiles in each group. We have also included bar graphs with each donor highlighted (Supp Figure 3D).

c. It is unclear how the genes in 1E and F were selected. What are the DEG identified from the single cell RNA seq data when obese and lean patients are compared for each identified subcluster?

Given the low frequency of cells in certain monocyte clusters, we pooled all three clusters to determine overall differences in their transcriptomes with maternal obesity. Genes highlighted in Figures 1E and 1F are the top candidates in that analysis based on their fold change relative to leans.

As stated above, this paper is focused on the impact of maternal obesity on fetal monocytes and as such group comparison for the additional clusters on the UMAP is beyond the scope of this manuscript. However, all sequencing data is available for the scientific community to conduct additional analyses.

d. The module scores in 1G should be plotted on the actual single cell map to verify that these signatures are only obtained/different in Monocyte populations and are not driven by any other cell type profiled.

Supp Figure 3E provides a feature plot for module score of each cell. As seen in the figure, these pathways are primarily observed in monocytes (the classical and cytokine high subsets).

2. A comparison between the viral and *E. coli* responses is missing. A commentary for the similarity/disparity between these responses could be of potential interest.

We thank the reviewer for the suggestion. We have compared overall transcriptional responses with *E. coli* and RSV responses is summarized in Supp Figures 5F and 5G.

3. Epigenetic data are presented in non-standard manner that makes interpretation of data quality and conclusion difficult to interpret.a. Average profiles of ATAC-seq signals around the TSS for the two groups should be performed following depth normalization to ensure that the profiles look similar and are comparable between groups. The signal at differential sites should also be shown and replicates should be indicated to ensure robustness of the results. What are the identified motifs of transcription factors associated with the differential regions – Is NfKb identified given that a significant role is attributed in figure 5.

We thank the reviewer for their useful comments. Unfortunately, due to low starting cell numbers, the numbers of peaks from each biological replicate were low. Therefore, we pooled the samples by group and performed differential analyses using HOMER. We however provide average TSS enrichment with a +/- 3KB window around the promoter for each group in Supp Figure 6C. Based on the reviewer’s suggestion, we performed motif analysis on differential peaks, and these results are shown in Figure 5G.

b. Histone Elisa and Histone Flow are not the golden standard to indicate epigenetic/chromatin changes. ChIP (PCR or Seq) is required to be performed to identify (1) locations in which changes occur and (2) identify the affected histone mark. It is unclear what is the interpretation of the differences observed in H3K9me3 -a repressive mark that is predominantly distributed in pericentromeric heterochromatin.

While we recognize that Histone flow and ELISA are not standard assays for measuring global changes in histone methylation or acetylation, these approaches are well accepted in human studies when cell numbers are limited (Wimmers et al. 2021; Arunachalam et al. 2020). Given the significant differences observed at baseline, we performed ChIP-seq (CUT&Tag given the low cell numbers) on H3K9me3- enriched regions of the genome. These data are now presented in Figure 4H-4J. We recognize that H3K9me3 marks heterochromatin in resting cells, but this modification also marks developmental genes poised to respond to subsequent cues from the environment. More importantly, following LPS stimulation and downstream TLR signaling, we expect a global loss of this mark especially within enhancer regions resulting in enhanced accessibility of a subset of NF-κB genes (Zhu, van Essen, and Saccani 2012). Our data demonstrates that the loss of H3K9me3 mark following LPS stimulation is less pronounced with maternal obesity.

4. The PCA plots in Figure6 are uninterpretable. What is the meaning of the circles in figure 6C and D and how are were they obtained? They do not appear to support PCA grouping.

Based on recommendations from all reviewers, we have removed PCAs from Figure 6.

Reviewer #3 (Recommendations for the authors):– Since the authors are working with frozen cells, analysis of viability after thawing is essential. There is no mention of viability staining or measurement after thawing anywhere.Please provide, otherwise this is a major confounder.

We understand that cell viability is a general concern with cryo-prerserved samples. We routinely assess viability during thaws and have so far not observed any differences between the two groups. We have included a representative graph comparing proportions of viable cells, measured using flow cytometry in Supp Figure 2H

– Mode of delivery is a major confounder which can affect the CBMC response. Did the authors check whether this affected their results?

Mode of delivery has been previously shown to alter cord blood cytokine profile (IL-13 and IFNγ) and alter Th2 responses ex vivo (Ly et al. 2006). We made a conscious effort of including an equal mix of UCB samples from C-section and vaginally delivered babies in all experiments. We also provide a panel in Supp Figure 2 that shows no differences in the ex vivo monocyte responses to. LPS based on mode of delivery.

– Could the authors provide the statistics on table 1? Where are the two groups different?

We have added group statistics on table 1.

– Please provide a gating strategy.– The monocyte subset analysis by flow cytometry is not according to recommendations (3 subsets: classical CD14++CD16dim, intermediate CD14++CD16+, non-classical CD14dimCD16++). Why was this chosen and could the authors go back and provide the recommended monocyte classifications?

The three monocyte subsets are difficult to characterize in cord blood, compared to adult human blood. However, we have reanalyzed data by biplots of CD14 and CD16 rather than histograms of CD16 on CD14+HLA-DR+ monocytes and provide an example of our gating strategy is provided in Supp Figure 2.

– Why was ELISA used to determine histone marks? This is very non-specific and only tells you something about the global H3K4 methylation, not gene specific. ChIP-PCR is a relatively easy method and doesn't require expensive sequencing if you already have ATAC-seq data

We initially wanted to characterize global changes in histone modifications, in a high throughput manner. Histone ELISA and Histone flow cytometry are routinely used to assess the global landscape using a small number of cells in the human immunology field. However, given the significant difference in H3K9me3 profile, we performed ChIP-seq, as recommended by 2 of the reviewers, and identified gene specific differences between the groups (Figures 4H-J).

– Was testing for equal variance performed?– Outliers were taken out before normality testing. However both ROUT and Grubbs assume a gaussian distribution. Please consider doing the normality tests first and only take out outliers if data are normally distributed.

We apologize for the erroneous reporting in Methods. Tests for normality, equal variance (F-test) and then outlier testing was done on all experiments with large sample sizes – plasma cytokine, immune phenotyping, and ex vivo responses. Experiments with smaller sample sizes were tested for normality and two-way differences (t-test). Statistical analyses section in Methods have been updated to reflect this.

– Whereas almost the whole manuscript is very well written, the discussion lacks in many things. First of all, the description of the findings is too long, but the comparison to other literature is very short. There is no mention of limitations at all, I cannot imagine the authors cannot think of limitations. And what about future perspectives? How can these findings influence the clinics or future pregnancies? I think the findings are important, so the authors are hereby invited to tell the readers too.

We thank the reviewer for their useful suggestion. We have edited the discussion, reorganized citations of research leading up to this study. Finally, we’ve added a section on limitations, future directions, and potential applications of our findings in the clinical setting.

References

Arunachalam, Prabhu S., Florian Wimmers, Chris Ka Pun Mok, Ranawaka A. P. M. Perera, Madeleine Scott, Thomas Hagan, Natalia Sigal, et al. 2020. “Systems Biological Assessment of Immunity to Mild versus Severe COVID-19 Infection in Humans.” *Science* 369 (6508): 1210–20.

Ly, Ngoc P., Begoña Ruiz-Pérez, Andrew B. Onderdonk, Arthur O. Tzianabos, Augusto A. Litonjua, Catherine Liang, Daniel Laskey, et al. 2006. “Mode of Delivery and Cord Blood Cytokines: A Birth Cohort Study.” *Clinical and Molecular Allergy: CMA* 4 (September): 13.

Sureshchandra, Suhas, Norma Mendoza, Allen Jankeel, Randall M. Wilson, Nicole E. Marshall, and Ilhem Messaoudi. 2021. “Phenotypic and Epigenetic Adaptations of Cord Blood CD4^+^ T Cells to Maternal Obesity.” *Frontiers in Immunology* 12 (April): 617592.

Sureshchandra, S., R. M. Wilson, M. Rais, N. E. Marshall, J. Q. Purnell, K. L. Thornburg, and I. Messaoudi. 2017. “Maternal Pregravid Obesity Remodels the DNA Methylation Landscape of Cord Blood Monocytes Disrupting Their Inflammatory Program.” *Journal of Immunology* 199 (8): 2729–44.

Wilson, R. M., N. E. Marshall, D. R. Jeske, J. Q. Purnell, K. Thornburg, and I. Messaoudi. 2015. “Maternal Obesity Alters Immune Cell Frequencies and Responses in Umbilical Cord Blood Samples.” *Pediatric Allergy and Immunology: Official Publication of the European Society of Pediatric Allergy and Immunology* 26 (4): 344–51.

Wimmers, Florian, Michele Donato, Alex Kuo, Tal Ashuach, Shakti Gupta, Chunfeng Li, Mai Dvorak, et al. 2021. “The Single-Cell Epigenomic and Transcriptional Landscape of Immunity to Influenza Vaccination.” *Cell* 184 (15): 3915–35.e21.

Zhu, Yina, Dominic van Essen, and Simona Saccani. 2012. “Cell-Type-Specific Control of Enhancer Activity by H3K9 Trimethylation.” *Molecular Cell* 46 (4): 408–23.